# The challenge of mapping the human connectome based on diffusion tractography

Klaus H. Maier-Hein

Tractography based on non-invasive diffusion imaging is central to the study of human brain connectivity. To date, the approach has not been systematically validated in ground truth studies. Based on a simulated human brain data set with ground truth tracts, we organized an open international tractography challenge, which resulted in 96 distinct submissions from 20 research groups. Here, we report the encouraging finding that most state-of-the-art algorithms produce tractograms containing 90% of the ground truth bundles (to at least some extent). However, the same tractograms contain many more invalid than valid bundles, and half of these invalid bundles occur systematically across research groups. Taken together, our results demonstrate and confirm fundamental ambiguities inherent in tract reconstruction based on orientation information alone, which need to be considered when interpreting tractography and connectivity results. Our approach provides a novel framework for estimating reliability of tractography and encourages innovation to address its current limitations.

#A full list of authors and their affliations appears at the end of the paper

Tractography, a computational reconstruction method based on diffusion-weighted magnetic resonance imaging (DWI), attempts to reveal the trajectories of white matter pathways in vivo and to infer the underlying structural connectome of the human brain[1]. Numerous algorithms for tractography have been developed and applied to connectome research in the field of neuroscience[2] and psychiatry[3]. Given the broad interest in this approach, advantages and shortcomings of tractography have been addressed using a wide range of approaches[1, 4–8]. Particularly, in vivo tractography of the human brain has been evaluated by subjective assessment of plausibility[9, 10] or qualitative visual agreement with post-mortem Klingler-like dissections[11, 12]. Reproducibility[13] or data prediction errors[14–16] have been evaluated in the context of tractography model verification. However, these evaluations cannot validate the accuracy of reconstructions due to the lack of ground truth information[17]. Ex vivo imaging and tracing[17–23] or physically[24–30] and numerically simulated phantoms[31–34] allow validation to some extent, and in specific circumstances such as basic fiber configurations. The nervous system, however, is complex and precise ground truth information on the trajectories of pathways and their origins, as well as terminations in the human brain is lacking. This makes it hard to obtain quantitative and comprehensive reliability estimations of tractography and to determine which discoveries are reliable when regarding brain connectivity in health and disease.

State-of-the-art tractography algorithms are driven by local orientation fields estimated from DWI, representing the local tangent direction to the white matter tract of interest[1].

Conceptually, the principle of inferring connectivity from local orientation fields can lead to problems as soon as pathways overlap, cross, branch, and have complex geometries[7, 35, 36]. Since the invention of diffusion tractography, these problems have been discussed in schematic representations or theoretical arguments[7, 8, 37], but have not yet been quantified in brain imaging. To determine the current state of the art in tractography, we organized an international tractography competition (tractometer.org/ismrm_2015_challenge). We employed simulated DWI of a brain-like geometry as a novel reliability estimation method that allowed for a quantitative evaluation of the submissions based on the Tractometer connectivity metrics[38].

At the closing of the competition, we evaluated 96 distinct tractography pipelines submitted by 20 different research groups, in order to assess how well the algorithms were able to reproduce the known connectivity. We also assessed essential processing steps to pinpoint critical flaws that many current pipelines have in common. An important positive finding is that most proposed algorithms are able to produce tractograms containing 90% of the ground truth bundles, recovering about one-third of their volumetric extent. At the same time, most algorithms produce large amounts of false-positive bundles, even though they are not part of the ground truth. Results do not improve when employing higher-quality data or even using the gold standard field of local tract orientations at high spatial resolution. The findings highlight that novel technological and conceptual developments are needed to address these limitations.

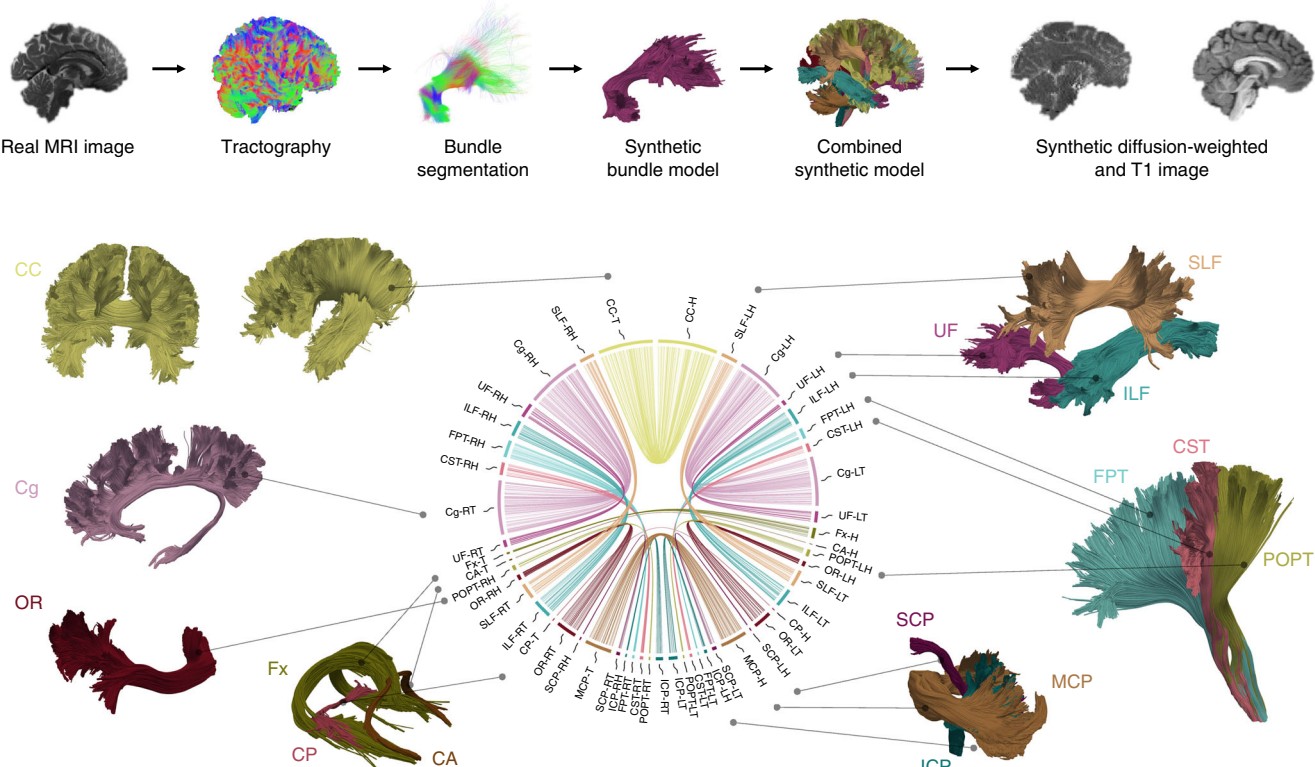

**Fig. 1** Overview of synthetic data set. The top row summarizes the phantom generation process. The simulated images are generated from 25 major bundles, which are shown in the bottom part of the figure. These were manually segmented from a whole-brain tractogram of a HCP subject and include the CC, cingulum (Cg), fornix (Fx), anterior commissure (CA), optic radiation (OR), posterior commissure (CP), inferior cerebellar peduncle (ICP), middle cerebellar peduncle (MCP), superior cerebellar peduncle (SCP), parieto-occipital pontine tract (POPT), cortico-spinal tract (CST), frontopontine tracts (FPT), ILF, UF, and SLF. The connectivity plot in the middle shows the phantom design. The segment positions correspond to the involved endpoint region (from top to bottom: frontal lobe, temporal lobe, parietal lobe, occipital lobe, subcortical region, cerebellum, brain stem). The radial segment length and the connection number in the plot are chosen according to the volume of the respective bundle endpoint region. Abbreviations: right (R) and left (L) hemisphere, head (H) and tail (T) of each respective bundle

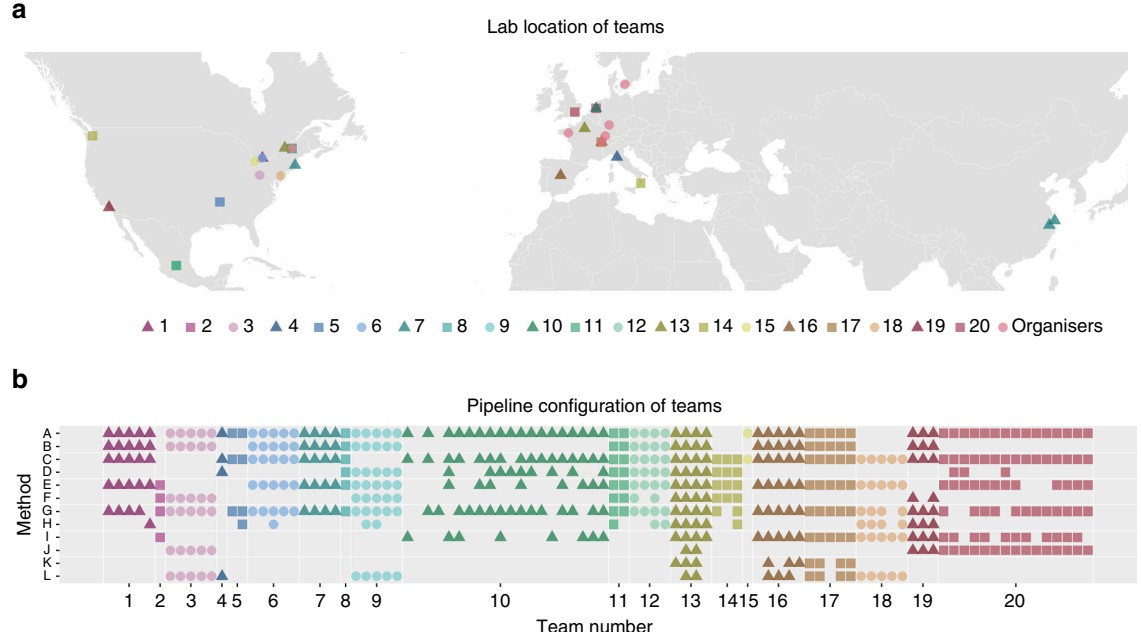

**Fig. 2** Summary of teams and tractography pipeline setups. **a** Location of the teams' affiliated labs. **b** Configuration of the different pipelines used for processing (A: motion correction, B: rotation of b-vectors, C: distortion correction, D: spike correction, E: denoising, F: upsampling, G: diffusion model beyond diffusion tensor imaging (DTI), H: tractography beyond deterministic, I: anatomical priors, J: streamline filtering, K: advanced streamline filtering, L: streamline clustering)

## Results

**Data sets and submissions.** Prior investigations of tractography methodology have chosen artificial fiber geometries to construct synthetic ground truth models[26, 38]. Here, we defined our models based on the fiber bundle geometry of a high-quality Human Connectome Project (HCP) data set that was constructed from multiple whole-brain global tractography maps[39] (Fig. 1). Following the concepts introduced in ref. [40], an expert radiologist (B.S.) extracted 25 major tracts (i.e., bundles of streamlines) from the tractogram. This ground truth data set included association, projection, and commissural tracts that have been previously described using post-mortem anatomical and electrophysiological methods[41]. In total the tracts occupy 71% of the white matter volume in the human brain. The data set features a brain-like macro-structure of long-range connections, mimicking in vivo DWI clinical-like acquisitions based on a simulated diffusion signal. An additional anatomical image with T1-like contrast was simulated as a reference. The final data sets and all files necessary to perform the simulation are openly available (see Data availability).

Twenty research groups with extensive expertise in diffusion imaging from 12 countries (Fig. 2a) participated in the competition and submitted a total of 96 tractograms (see Data availability) generated using a large variety of tractography pipelines with different pre-processing, local reconstruction, tractography, and post-processing algorithms (Fig. 2b, Supplementary Note 1).

**Performance metrics and evaluation.** The Tractometer connectivity metrics[38] were used for a quantitative evaluation of the submissions. Based on the known ground truth bundles, we calculated true positives, corresponding to the valid connection (VC) ratio, that is, the proportion of streamlines connecting valid end points and the associated number of valid bundles (VB), where a bundle is a group of streamlines. We also computed false positives, corresponding to the invalid connection (IC) ratio and the associated number of invalid bundles (IB), as well as reconstructed volumes, based on the bundle volumetric overlap (OL) and volumetric overreach (OR) in percent (see "Methods" section for details and Supplementary Figs. 1, 2 for alternative metrics).

**Tractograms contained most of the ground truth bundles.** The volumetric reconstruction of the existing bundles varied greatly from tract to tract. Figure 3a shows that identified VBs can be arbitrarily grouped into three clusters of very hard, hard, and medium difficulty, according to the percentage of OL. Figure 3b shows corresponding examples that were reconstructed by different tractography techniques. All submissions had difficulties reconstructing the smallest tracts, that is, the anterior (CA) and posterior commissures (CP) that have a cross-sectional diameter of no more than 2 mm, or one or two voxels (very hard, 0% <= OL < 10%). A family of hard bundles was partly recovered (10% <= OL < 50%). Bundles of medium difficulty were the corpus callosum (CC), inferior longitudinal fasciculus (ILF), superior longitudinal fasciculus (SLF), and uncinate fasciculus (UF) with an average of more than 50% volumetric recovery (50% <= OL <= 100%). A Pearson product-moment correlation coefficient was computed to assess the relationship between OL and OR ($r = 0.88$, $p < 10^{-8}$), indicating a direct link between the probability of reconstructing a greater portion of a tract (OL) and generating artefactual trajectories (OR).

Figure 4 shows that on average 21 out of 25 VBs (median 23) were identified by the participating teams with only four teams submitting tractograms that contained an OL of more than 60%. No submission contained all 25 VBs, but 10 submissions (10.4%) recovered 24 VBs, and 69 submissions (71.9%) detected 23 or more VBs (Fig. 5a). However, tractography pipelines clearly need to improve their recovery of the full spatial extent of bundles: the mean value of bundle volume overlap (OL) across all submissions was $36 \pm 16\%$, with an average overreach (OR) of $29 \pm 26\%$ (Fig. 4c). At the level of individual streamlines, an average of $54 \pm 23\%$ connections were valid (Fig. 4a).

**Tractograms contained more invalid than valid bundles.** Across submissions, $36 \pm 17\%$ of the reconstructed individual streamlines connected regions that were not actually connected. The fraction of streamlines not connecting any endpoints was $10 \pm 15\%$. Even though not part of the ground truth, these streamlines often occur in dense, structured, and coherent bundles. Submitted tractograms

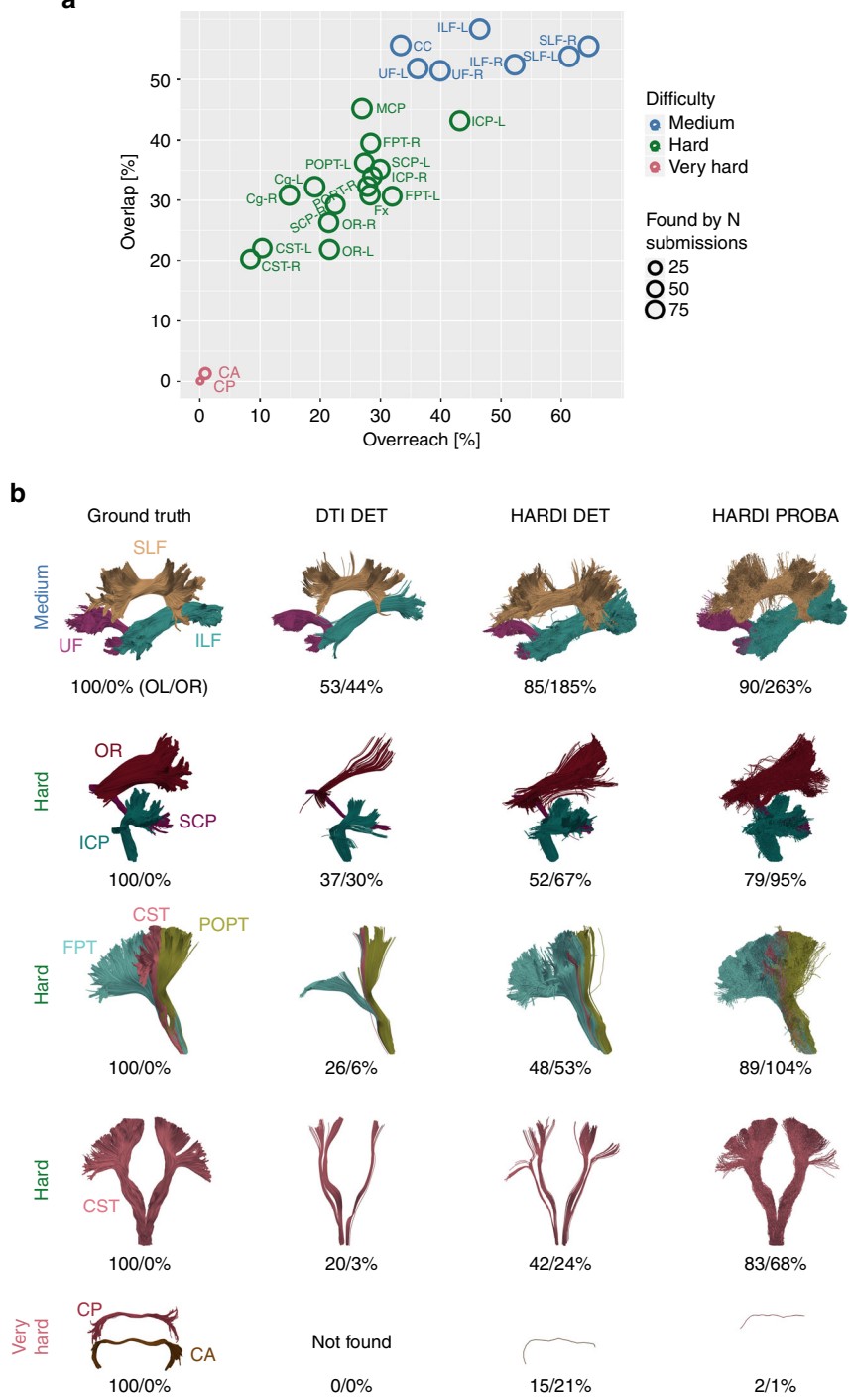

**Fig. 3** Tractography identifies most of the ground truth bundles, but not their full extent. **a** Overview of scores reached for different bundles in ground truth. Average overlap (OL) and average overreach (OR) scores for the submissions (red: very hard, green: hard, blue: medium, for abbreviations see Fig. 1). **b** Representative bundles for DTI deterministic (DET) tracking come from submission 6/team 20, high angular resolution diffusion imaging (HARDI) deterministic tracking from submission 0/team 9, and HARDI probabilistic (PROBA) tracking from submission 2/team 12 (see Supplementary Note 5 for a discussion of these submissions). The first column shows ground truth VBs for reference. The reported OL and OR scores correspond to the highest OL score reached within the respective class of algorithms

contained an average of 88 ± 58 IBs, which is more than four times the amount of VBs they contained on average (Fig. 4b). This demonstrates the inability of current state-of-the-art tractography algorithms to control for false positives. Forty-one of these IBs occurred in the majority of submissions (Fig. 5, Supplementary Fig. 3). Overall average precision on the bundle level was 23 ± 9% (recall 85 ± 15%, specificity 93 ± 5%). Submissions with at least 23

VBs showed no fewer than 37 IBs (mean 88 ± 39, $n = 69$). Submissions with 23 or more VBs and a volumetric bundle overlap of >50% identified between 99 and 204 IBs.

The bundles illustrated in Fig. 5b were systematically found by 81–95% of submissions without being part of the ground truth. Interestingly, several of these invalid streamline clusters exhibited similarities in anatomical location to bundles known or

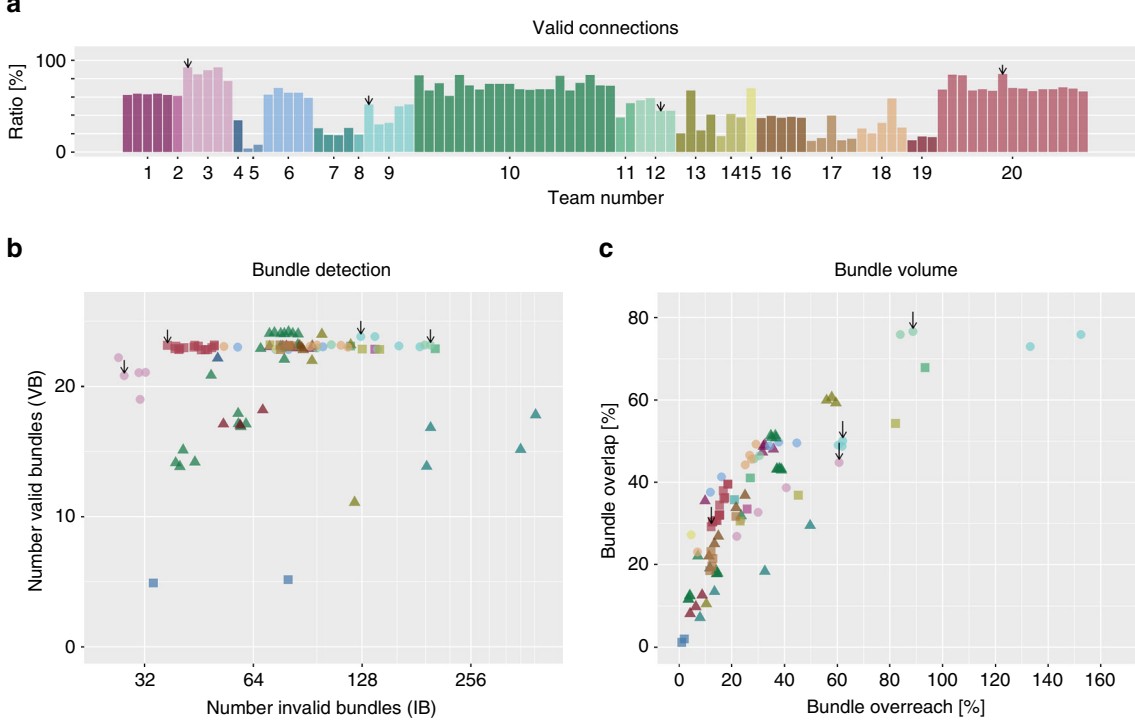

**Fig. 4** Between-group differences in tractography reconstructions of VBs and IBs. Overview of the scores reached by the different teams as **a** percentage of streamlines connecting valid regions, **b** number of detected VBs and IBs (data points are jittered to improve legibility), and **c** volume overlap (OL) and overreach (OR) scores averaged over bundles. Black arrows mark submissions used in the following figures (see Supplementary Note 5 for discussion)

previously debated in tractography literature, such as the frontal aslant tract (FAT)[42], the arcuate fasciculus (AF)[43], the inferior frontal occipital fasciculus (IFOF)[44], the middle longitudinal fasciculus (MdLF)[45], the extreme capsule fasciculus[46], the superior fronto-occipital fasciculus (SFOF)[44, 47], and the vertical occipital fasciculus (VOF)[48]. These findings suggest that evidence for the existence of tracts should not be taken solely from tractography at its current state but complemented by other anatomical and electrophysiological methods.

**Higher image quality may improve tractography validity.** To confirm that our findings revealed fundamental properties of tractography itself and are not related to effects of our specific phantom simulation process, we ran two independent implementations of deterministic streamline tractography (Supplementary Note 2) directly on the ground truth field of fiber orientations (Fig. 6), that is, without using the diffusion-weighted data at all. This experiment was repeated for multiple resolutions (2, 1.75, 1.5, 1.25, 1.0, 0.75, and 0.5 mm). This setup was, thus, independent of image quality, artifacts, and many other influences from specific pipeline configurations and the phantom generation process. Based on the ground truth orientations, the tractography pipelines achieved overlap scores ($76 \pm 6\%$) that were previously unreached at similar levels of overreach ($29 \pm 8\%$). VC ratios were between 71 and 82%. However, the tractograms still contained $102 \pm 24$ IBs (minimum 73).

**Methodological innovation may improve tractography validity.** Our results show that the geometry of many junctions in the simulated data set is too complex to be resolved by current tractography algorithms, even when given a perfect ground truth field of orientations. Thus, the problems seem to relate to essential ambiguities in the directional information (Fig. 7). They persisted in supplementary experiments performed to test the potential of

currently available anatomical constraints and global tractography approaches (Supplementary Note 2), in which none of the additionally ran methods surpassed the challenge submissions in bundle detection performance (Supplementary Fig. 4).

We further investigated the ambiguities tractography encounters in the synthetic phantom as well as in an in vivo data set. In the temporal lobe, for example, multiple bundles overlap and clearly outnumber the count of fiber orientations in most of the voxels. As illustrated in Fig. 8, single fiber directions in the diffusion signal regularly represent multiple bundles (see also Supplementary Movie 1). Such funnels embody hard bottlenecks for tractography, leading to massive combinatorial possibilities of plausible configurations for connecting the associated bundle endpoints as sketched in Figs. 7c and 8c. Consequently, for the real data set as well as the synthetic phantom, dozens of structured and coherent bundles pass through this bottleneck, exhibiting similar fiber counts (cf. Supplementary Figs. 5, 6) and a wide range of anatomically reasonable geometries as illustrated in Supplementary Movie 2. A tractogram based on real HCP data exhibits a whole family of theoretically plausible bundles going through the temporal lobe bottleneck even though, locally, the diffusion signal often shows only one fiber direction (cf. Fig. 8d). Methodological innovation will be necessary to resolve these issues and better exploit additional information sources that complement the local orientation fields estimated from DWI.

**Statistical analysis of processing steps.** Effects of the methodological setup of the different submissions on the results were investigated in a multivariable linear mixed model and revealed the influence of the individual processing steps on the tractography outcome (Table 1). The choice of tractography algorithm, as well as the post-tracking filtering strategy and the underlying diffusion modeling had a strong effect on overall scores, revealing a clear tradeoff between sensitivity and specificity (Supplementary Note 3). Manual editing of tractograms following anatomical priors had a

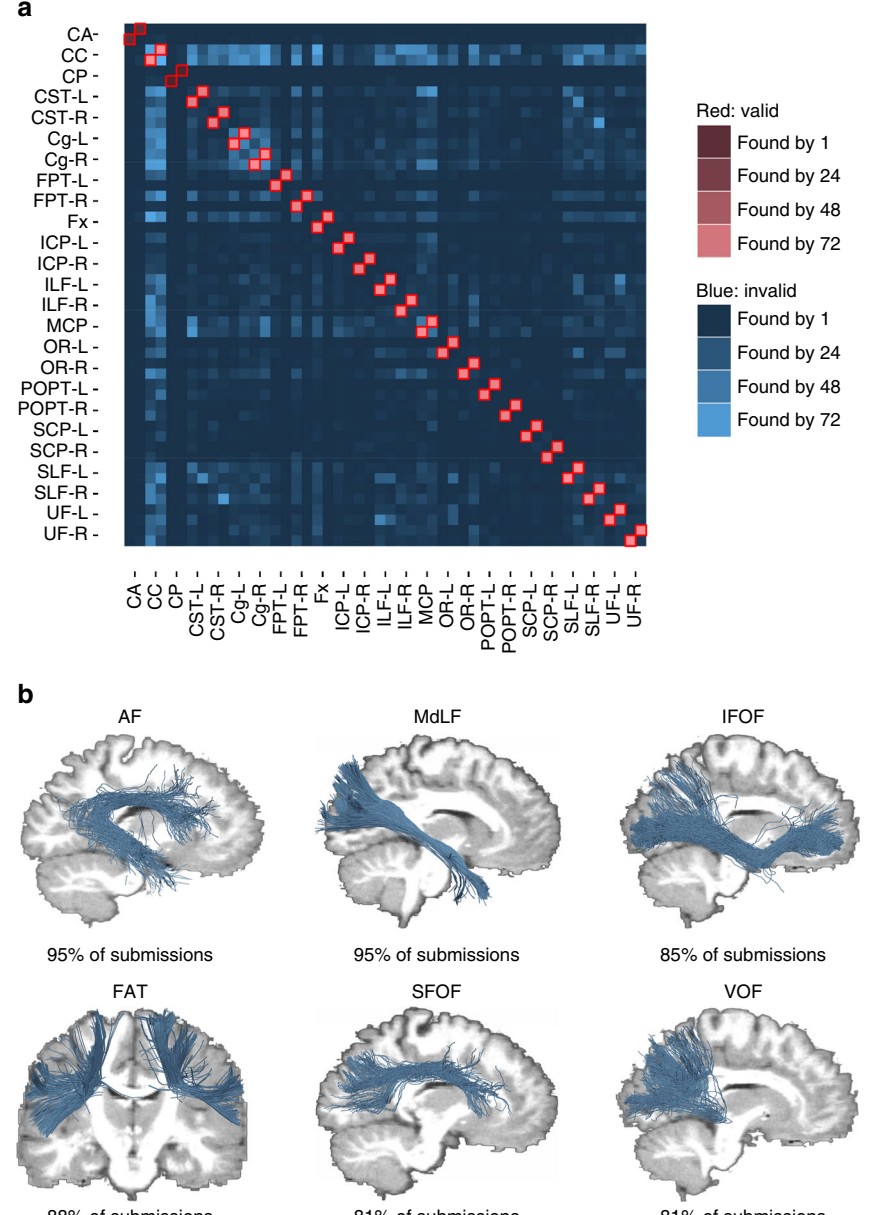

**Fig. 5** Overview of VBs and IBs and examples of invalid streamline clusters. **a** Each entry in the connectivity matrix indicates the number of submissions that have identified the respective bundle. The two rows and columns of each bundle represent the head-endpoint and tail-endpoint regions. The connectivity matrix indicates a high number of existing tracts that were identified by most submissions (red). It also indicates systematic artefactual reconstructions across teams (blue). **b** Examples of IBs that have been consistently identified by more than 80% of the submissions, but do not exist in the ground truth data set. The AF, for example, was generated from ILF and SLF crossing streamlines, whereas the IFOF was generated from by crossing ILF and UF streamlines. The MdLF, FAT, SFOF, and VOF were other examples of highly represented IBs

negative impact on the number of VBs identified (mean effect: 3.8 $\pm$ 2.6 bundles) and on the bundle overlap (mean effect: $-15 \pm 11\%$). However, such techniques showed a positive impact on the average bundle overreach (mean effect: $-16 \pm 9\%$). Notably, Team 3 post-processed the tractograms using clustering, reaching 92% validly connecting streamlines keeping only the larger clusters.

## Discussion

We assessed current state-of-the-art fiber tractography approaches using a ground truth data set of white matter tracts and connectivity that is representative of the challenges that may occur in human in vivo brain imaging. Advanced tractography strategies in combination with current diffusion modeling

techniques successfully recovered most VBs, covering up to 77% of their volumetric extent. This result demonstrates the capability of current methods and teams to adequately handle numerous artifacts in DWI and overcome local crossing situations during tract reconstruction. However, tractography also produced thick and dense bundles of plausible looking streamlines in locations where such streamlines did not actually exist. When focusing on the 64 bundles that were systematically recovered by the majority of submissions, 64% of them were in fact absent from the ground truth. Current tractography pipelines, and even tracking of the ground truth fiber orientations on high-resolution images, produce substantial amounts of false-positive bundles. The employed simulation-based approach cannot quantify the effects related to in vivo connectivity in an absolute sense; that is, our results do

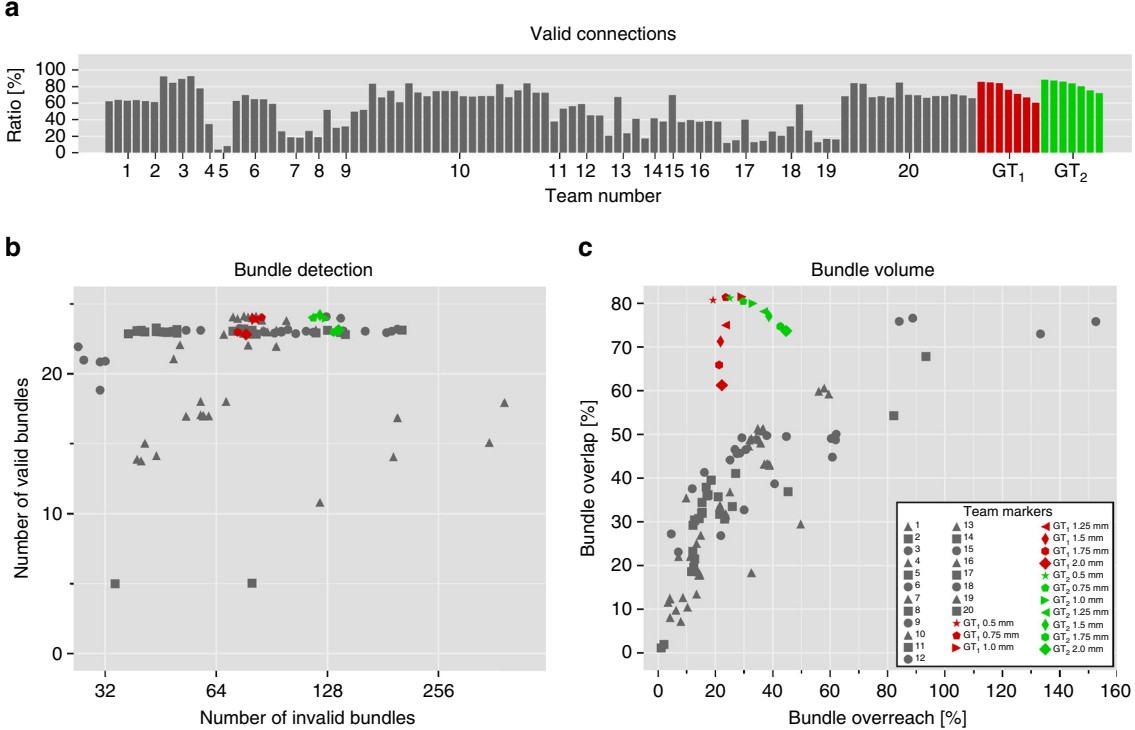

**Fig. 6** Tractography on ground truth directions with no noise still affected by IB problem. We applied deterministic tractography directly to the ground truth vector field with multiple resolutions (2, 1.75, 1.5, 1.25, 1.0, 0.75, and 0.5 mm). Two independent implementations of deterministic tractography methods were used to obtain the results (GT₁ and GT₂, cf. Supplementary Note 2). **a** Percentage of streamlines connecting valid regions. **b** Number of detected VBs and IBs (data points are jittered to improve legibility). **c** Volume overlap and overreach scores averaged over bundles

not mean that anyone who is doing tractography should expect the reported VB-to-IB and coverage-to-overreach ratios. However, the presented findings do expose the degree of ambiguity associated with whole-brain tractography and show that the computational problem of tractography goes far beyond the local reconstruction of fiber directions[1, 7] and issues of data quality. Our findings, therefore, present a core and open challenge for the field of tractography and connectivity mapping in general.

Previous studies have reported high invalid-connection ratios under simplified conditions[26, 38] (www.tractometer.org), and some of the underlying ambiguities in tractography have been discussed using schematic representations and theoretical arguments[1, 7, 8, 37]. Regions of white matter bottlenecks have been discussed in the past[35] and have been highlighted as critical with respect to tracto-graphic findings[36]. The current results reveal the consequences of such limitations under more complex conditions as might be found in human brain studies in vivo, addressing important questions that previously remained speculative. The findings were derived from a brain-like geometry that encompasses some of the major known long-range connections and covers 71% of the white matter. Future versions of the phantom are planned to include additional bundles such as the middle and inferior temporal projections of the AF, the MdLF, and the IFOF, as well as smaller U-fibers, medial forebrain fibers, deep nuclei, and connections between them. In addition, more advanced diffusion modeling methods will allow generating even more realistic DWI signals, potentially simulated at increased spatial and q-space resolutions[49].

These developments, however, will not resolve the fundamental ambiguities which tractography faces and thus will only have a limited effect on the main findings of our study. We showed that false-positive bundles occur at similar rates even when using the maximal angular precision of the signal, that is, using ground truth orientations. These findings confirm those shown in previous studies[5] and relate to the fundamental problem formulation

in tractography: inferring connectivity from local orientation fields. Increasing the anatomic complexity of the phantom by adding more bundles most likely will even lead to further increased false-positive rates. The construction process of the current phantom resembles a potential limitation, since it involves tractography itself and thus raises self-validation issues. This aspect should be considered in direct method comparisons as there may exist a possible bias toward algorithms that are similar to the algorithm used for phantom generation. This caveat, however, has only a very limited effect on our general findings. It can be expected that the identified limitations of tractography will become even more pronounced in phantoms of higher anatomic complexity that might be achievable by involving independent methods such as polarized light imaging[50]. In summary, our observations confirm the fundamental ill-posed nature of the computational problem that current tractography approaches strive to solve.

Accordingly, substantial methodological innovations will be necessary to resolve the problem of IBs. Several directions of current research might improve the specificity of tractography. Streamline filtering techniques can optimize the signal prediction error in order to reduce tractography biases[14, 16, 51]. They are part of the more general trend to integrate non-local information, as well as advanced diffusion microstructure modeling that goes beyond the raw directional vectors[52–58]. Recent advances in machine-learning-driven tractography also show great potential in improving the specificity of tractograms[59, 60]. Future versions of our phantom will be generated with multiple b-values, better signal-to-noise ratio (SNR), and fewer artifacts to further encourage research in these directions.

In addition, tractography should increasingly employ reliable anatomical priors from ex vivo histology, high-resolution post-mortem DWI[61], or complementary electrophysiology for optimal guidance. While manual or automated clean-up of streamlines

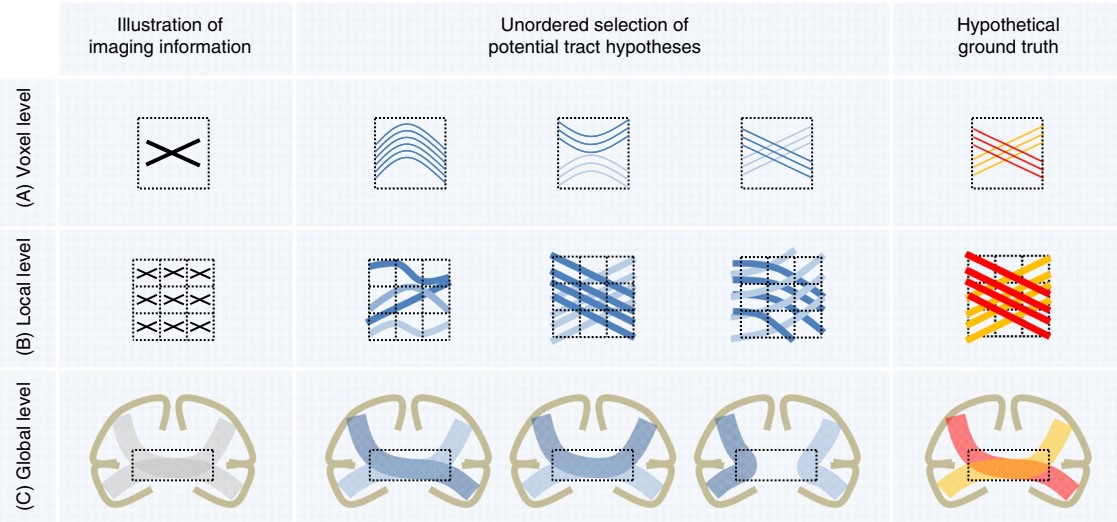

**Fig. 7** Ambiguous correspondences between diffusion directions and fiber geometry. The three illustrations at voxel, local, and global level are used as an example to illustrate the possible ambiguities contained in the diffusion imaging information that may lead to alternative tract reconstructions. (A) The intra-voxel crossing of fibers in the hypothetical ground truth leads to ambiguous imaging information at voxel level[7]. (B) Similarly, the imaging representation of local fiber crossings can be explained by several other configurations[7]. (C) At a global level, white matter regions that are shared by multiple bundles (so-called "bottlenecks", dotted rectangles)[35] can lead to many spurious tractographic reconstructions[36]. With only two bundles in the hypothetical ground truth (red and yellow bundle), four potential false-positive bundles emerge. Please note that the hypothetical ground truth used in the global-level example is anatomically incorrect as most of the callosal fibers are homotopically distributed (i.e., connect similar regions on both hemispheres)

may help (as demonstrated by our results showing decreased overreach at the expense of VB detection and volumetric reconstruction), the real challenge is our limited knowledge of the anatomy to be reconstructed. Currently, post-mortem dissection with Klingler's method reveals the macroscopic organization of the human brain white matter[11, 62–64], although this method shares some of the mentioned limitations of tractography in complex fiber configurations or near the cortex. In the future, the community will have to gain further insights into the underlying principles of white matter organization and increasingly learn how to leverage such information for tractography[1, 65, 66].

Potential advances achieved in tractography will have an important impact on graph-analytical studies of the structural connectome[2, 67]. The hitherto demonstrated diagnostic or predictive capability of such analyses (e.g., in psychiatric settings) should not let us overlook which aspects of tractography are reliable and which are not. One of the present findings is particularly relevant for the field of connectomics: the traditional metrics that require streamlines to exactly end in head or tail regions of a bundle are far too restrictive for bundle dissection and connectivity assessment. None of the submissions generated exact streamlines that perfectly overlap with ground truth bundles and dilated endpoint masks. This finding is in line with previous reports, which found termination of tracts in the gray matter (GM) to be inaccurate[5] and highlights an important limitation of approaches that use a voxel-wise definition of parcellations on the T1 image for selecting relevant streamlines. Future versions of our phantom will include ground-truth parcellations of the white matter/GM cortical band to encourage further developments for tackling these problems and extend the evaluation method to apply to graph theory metrics.

Despite any limitations, DWI is currently the only tool to map short and long-range structural brain connectivity in vivo and is essential for comparing brains, detecting differences, and simulating brain activity[39]. Our findings should foster the development of novel tractography methods that are carefully evaluated using the present approach. The most important goal for the next generation of tractography algorithms is an improved ability to reconstruct the full spatial extent of existing tracts while better controlling for false-positive connections. A tighter integration of anatomical priors, advanced diffusion microstructure modeling, and multi-modality imaging should help to resolve ambiguities in the signal and overcome current limitations of tractography[57, 58]. Fundamentally, there is an urgent need for methodological innovation in tractography in order to advance our knowledge of human white matter anatomy and build anatomically correct human connectomes[1, 7].

## Methods

**Generation of ground truth fiber bundles.** The set of ground truth long-range fiber bundles was designed to cover the whole human brain and features many of the relevant spatial configurations, such as crossing, kissing, twisting and fanning fibers, thus representing the morphology of the major known in vivo fiber bundles. The process to obtain these bundles consisted of three steps. First, a whole-brain global tractography was performed on a high-quality in vivo diffusion-weighted image. Then, 25 major long-range bundles were manually extracted from the resulting tractogram. In the third step, these bundles were refined to obtain smooth and well-defined bundles. Each of these steps is detailed in the following paragraphs.

We chose one of the diffusion-weighted data sets included in the Q3 data release of the HCP[39], subject 100307, to perform whole-brain global fiber tractography[52, 68]. Among other customizations, the HCP scanners are equipped with a set of high-end gradient coils, enabling diffusion encoding gradient strengths of $100\ \mathrm{mT\ m^{-1}}$. By comparison, most standard magnetic resonance scanners feature gradient strengths of about 30 to $40\ \mathrm{mT\ m^{-1}}$. This hardware setup allows the acquisition of data sets featuring exceptionally high resolutions (1.25 mm isotropic, 270 gradient directions) while maintaining an excellent SNR. All data sets were corrected for head motion, eddy currents and susceptibility distortions and are, in general, of very high quality[69–73]. Detailed information regarding the employed imaging protocols as well as the data sets themselves may be found on http://humanconnectome.org.

Global fiber tractography was performed using MITK Diffusion[74] with the following parameters: 900,000,000 iterations, a particle length of 1 mm, a particle width of 0.1 mm, and a particle weight of 0.002. Furthermore, we repeated the tractography six times and combined the resulting whole-brain tractograms into one large data set consisting of over five million streamlines. The selected parameters provided for a very high sensitivity of the tractography method. The specificity of the resulting tractogram was of lesser concern since the tracts of interest were extracted manually in the second step.

Bundle segmentation was performed by an expert radiologist using manually placed inclusion and exclusion regions of interest (ROI). We followed the concepts introduced in ref. [40] for the ROI placement and fiber extraction. Twenty-five

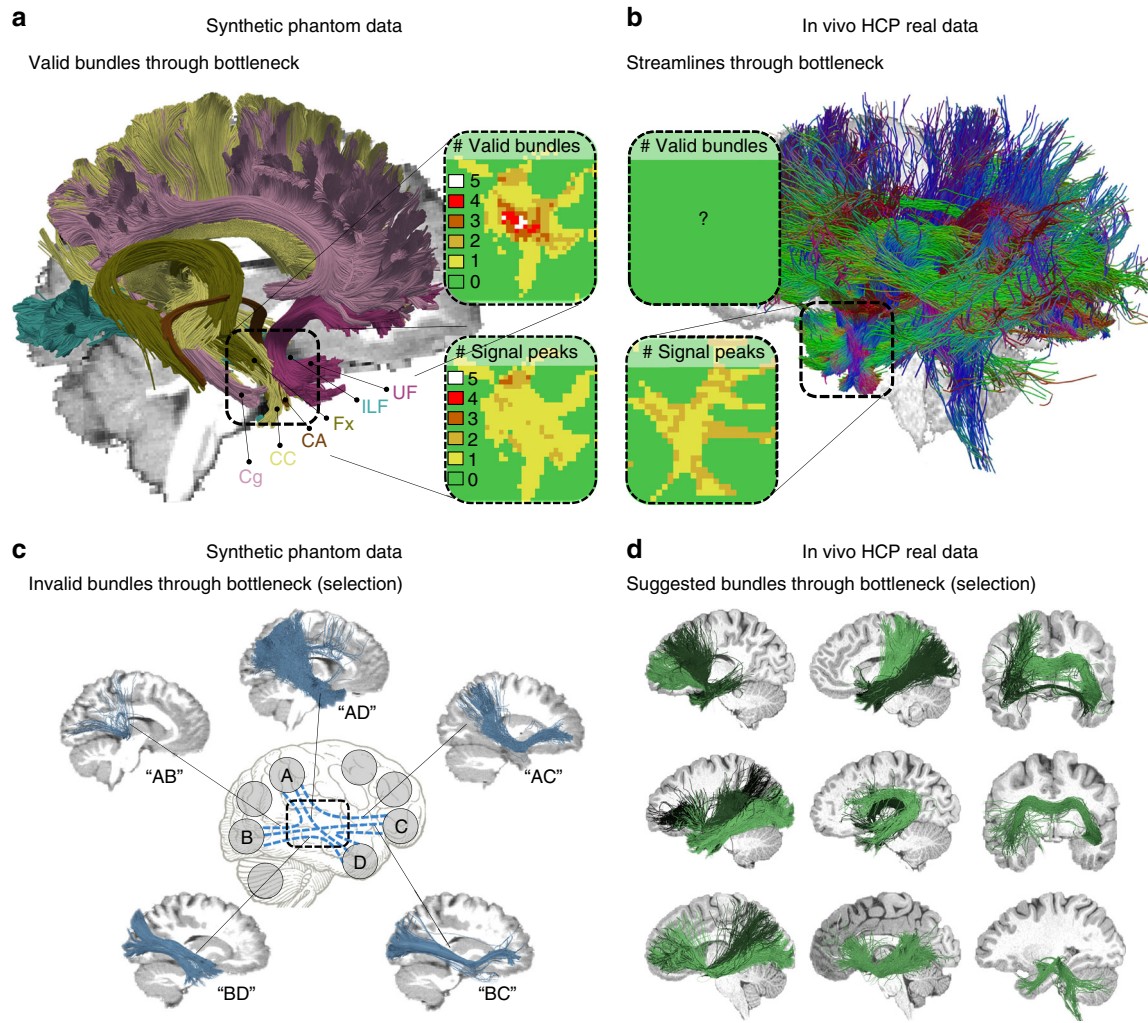

**Fig. 8** Bottlenecks and the fundamental ill-posed nature of tractography. **a** Visualization of six ground truth bundles converging into a nearly parallel funnel in the bottleneck region of the left temporal lobe (indicated by square region). The bundles per voxel (box "# Valid bundles") clearly outnumber the peak directions in the diffusion signal (box "# Signal peaks"). **b** Visualization of streamlines from a HCP in vivo tractogram passing through the same region. **c** Exemplary IBs that have been identified by more than 50% of the submissions, showing that tractography cannot differentiate between the high amount of plausible combinatorial possibilities connecting different endpoint regions (see Supplementary Movie 1). **d** Automatically QuickBundle-clustered streamlines from the in vivo tractogram going through the temporal ROI. The clustered bundles are illustrated in different shades of green. These clusters represent a mixture of true-positive and false-positive bundles going through that bottleneck area of the HCP data set (see Supplementary Movie 2)

bundles were extracted, covering association, projection, and commissural fibers across the whole brain (Fig. 1): CC, left and right cingulum (Cg), Fornix (Fx), anterior commissure (CA), left and right optic radiation (OR), posterior commissure (CP), left and right inferior cerebellar peduncle (ICP), middle cerebellar peduncle (MCP), left and right superior cerebellar peduncle (SCP), left and right parieto-occipital pontine tract (POPT), left and right cortico-spinal tract (CST), left and right frontopontine tracts (FPT), left and right ILF, left and right UF, and left and right SLF. As mentioned in the "Discussion" section, the IFOF, the MdLF, as well as the middle and inferior temporal projections of the AF were not included.

After manual extraction, the individual long-range bundles were further refined to serve as ground truth for the image simulation as also shown in Fig. 1. The original extracted tracts featured a large number of prematurely ending fibers and the individual streamlines were not smooth. To obtain smooth tracts without prematurely ending fibers, we simulated a diffusion-weighted image from each original tract individually using Fiberfox (www.mitk.org33). Since no complex fiber configurations, such as crossings, were present in the individual tract images and no artifacts were simulated, it was possible to obtain very smooth and complete tracts from these images with a simple tensor-based streamline tractography. Supplementary Fig. 7 illustrates the result of this refining procedure on the left CST.

**Simulation of phantom images with brain-like geometry**. The phantom diffusion-weighted images (Supplementary Movie 3) were simulated using Fiberfox

(www.mitk.org33), which is available as open-source software. We employed a four-compartment model of brain tissue (intra and inter-axonal), GM, and cerebrospinal fluid (CSF)[33]. The parameters for simulation of the four-compartment diffusion-weighted signal were chosen to obtain representative diffusion properties and image contrasts (compare[75] for details on the models). The intra-axonal compartment was simulated using the stick model with a T2 relaxation time of 110 ms and a diffusivity of $1.2 \times 10^{-9}$ m$^2$ s$^{-1}$. The inter-axonal compartment was simulated using the zeppelin model with a T2 relaxation time of 110 ms, an axial diffusivity of $1.2 \times 10^{-9}$ m$^2$ s$^{-1}$ and a radial diffusivity of $0.3 \times 10^{-9}$ m$^2$ s$^{-1}$. The GM compartment was simulated using the ball model with a T2 relaxation time of 80 ms and a diffusivity of $1.0 \times 10^{-9}$ m$^2$ s$^{-1}$. The CSF compartment was also simulated using the ball model with a T2 relaxation time of 2500 ms and a diffusivity of $2.0 \times 10^{-9}$ m$^2$ s$^{-1}$.

Using Fiberfox, one set of diffusion-weighted images and one T1-weighted image were simulated. The final data sets as well as all files needed to perform the simulation are available online (see Data availability).

The acquisition parameters that we report below were chosen to simulate images that are representative for a practical (e.g., clinical) setting, specifically a 5–10-min single shot echo-planar imaging scan with 2 mm isotropic voxels, 32 gradient directions, and a b-value of 1000 s mm$^{-2}$. The chosen acquisition setup represents a typical scenario for an applied tractography study and embodies a common denominator supported by the large majority of methods. Since acquisitions with higher b-values, more gradient directions and fewer artifacts are

**Table 1 Summary of the statistical analysis**

|  | VC | VB | IB | OL | OR |
|---|---|---|---|---|---|
| Motion correction | 12% ± 13% | 0.3 ± 2.8 | -3 ± 38.1 | 2% ± 12% | 7% ± 10% |
| Rotate b-vecs | -5% ± 14% | -1.2 ± 2.9 | 20.8 ± 39.6 | -6% ± 12% | -8% ± 9% |
| Distortion correction | -1% ± 13% | -2 ± 2.7 | -4 ± 36.8 | -2% ± 11% | -11% ± 10% |
| Spike correction | 1% ± 13% | 0.9 ± 2.6 | 4.6 ± 35.2 | 5% ± 11% | 10% ± 9% |
| Denoising | 1% ± 13% | 0.6 ± 2.6 | 3.2 ± 35.2 | 3% ± 11% | 12% ± 9% |
| Upsampling | 1% ± 13% | 0.2 ± 2.7 | 9.3 ± 36.4 | 4% ± 11% | 11% ± 9% |
| Beyond DTI | -12% ± 13% | 2.2 ± 2.6 | 21.5 ± 35.2 | 13% ± 11% | 15% ± 9% |
| Beyond deterministic | -10% ± 13% | -0.3 ± 2.6 | 41 ± 35.3 | 12% ± 11% | 17% ± 9% |
| Anatomical priors | 1% ± 13% | -3.8 ± 2.6 | -11.6 ± 35.2 | -15% ± 11% | -16% ± 9% |
| Streamline filtering | 21% ± 13% | 2.9 ± 2.8 | -41.6 ± 38.1 | 8% ± 12% | -4% ± 9% |
| Adv. streamline filtering | -11% ± 13% | 2.8 ± 2.7 | 0.9 ± 36.5 | 11% ± 11% | 4% ± 10% |
| Clustering | -3% ± 13% | 0.2 ± 2.7 | -1.1 ± 36.4 | -4% ± 11% | 2% ± 9% |

Green cells indicate a significant positive influence ($p < 0.05$) and red cells indicate a significant negative impact ($p < 0.05$). Numbers indicate the estimated mean effect on the metric and its standard deviation. The first column of the table represents the different parts of the processing pipeline that we have grouped into categories. The other columns represent the metrics: VC valid connections, VB valid bundles, IB invalid bundles, OL overlap, OR overreach

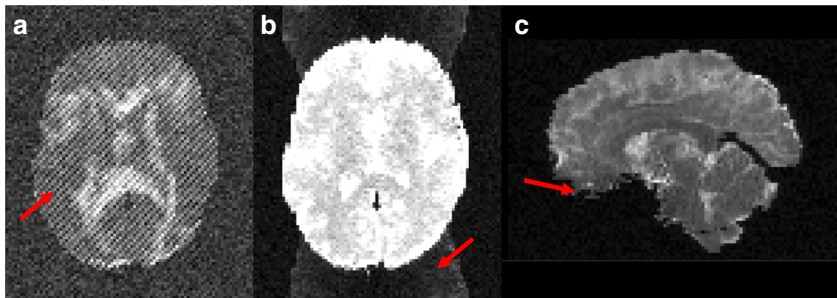

**Fig. 9** Illustration of artifacts included in the synthetic data set. Exemplary illustration of the spike (**a**), N/2 ghost (**b**), and distortion artifacts (**c**) contained in the final diffusion-weighted data set. Supplementary Movie 3 gives an impression of the complete synthetic data set provided

beneficial for tractography, we additionally report a least upper bound tractography performance under perfect image quality conditions using a data set that directly contains ground truth fiber orientation information at high spatial resolution with no artifacts (Fig. 6 and Supplementary Note 2).

The parameters are a matrix size of $90 \times 108 \times 90$, echo time (TE) 108 ms, dwell time 1 ms; $T2'$ relaxation time 50 ms. The simulation corresponded to a single-coil acquisition with constant coil sensitivity, no partial Fourier and no parallel imaging. Phase encoding was posterior-anterior. Two unweighted images with posterior-anterior/anterior-posterior phase encoding were also generated.

Since Fiberfox simulates the actual k-space acquisition, it was possible to introduce a number of common artifacts into the final image. Complex Gaussian noise was simulated yielding a final SNR relative to the mean white matter baseline signal of about 20. Ten spikes were distributed randomly throughout the image volumes (Fig. 9a). N/2 ghosts were simulated (Fig. 9b). Distortions caused by $B_0$ field inhomogeneities are introduced using an existing field map measured in a real acquisition and registered to the employed reference HCP data set (Fig. 9c). Head motion was introduced as random rotation ($\pm 4°$ around $z$-axis) and translation ($\pm 2$ mm along $x$-axis) in three randomly chosen volumes. Volume 6 was rotated by 3.36° and translated by −1.74 mm, volume 12 was rotated by 1.23° and translated by −0.72 mm, and volume 24 was rotated by −3.12° and translated by −1.55 mm.

The image with the T1-like contrast was generated at an isotropic resolution of 1 mm, an SNR of about 40 and no further artifacts as an anatomical reference.

**Performance metrics and evaluation**. The groups submitted sets of streamlines (see Data availability) and a brief description of their methods which is available in Supplementary Note 1. Potential operator-dependent errors were not taken into account but these are likely to have contributed to the quality of the final results. Probabilistic tractography techniques were preprocessed with a user-defined uncertainty threshold that each group decided independently before submission.

The Tractometer definition of a VC is extremely restrictive for current tractography algorithms, as it requires streamlines (1) not to exit the area of the ground truth bundle at any point and (2) to terminate exactly within the endpoint region that is defined by the dilated ground truth fiber endpoints (Supplementary Figs. 8, 9)[38]. Hence, we adopted an alternative definition with less stringent criteria

based on robust shape distance measures[76] and clustering between streamlines[77], as detailed in Supplementary Note 4. The bundle-specific thresholds were manually configured to account for bundle shape and proximity to other bundles. The following distances were used, with identical distances on both sides for lateralized bundles: 2 mm for CA and CP; 3 mm for CST and SCP; 5 mm for Cingulum; 6 mm for Fornix, ICP, OR, and UF; 7 mm for FPT, ILF, and POPT; 10 mm for CC, MCP, and SLF. The full script used to run this bundle recognition implementation was based on the DIPY library[78] (www.dipy.org) and is available online (Supplementary Software 1).

Once VCs are identified, the remaining streamlines can be classified into ICs and non-connecting streamlines. The details of this procedure are described in Supplementary Note 4. We clustered the remaining invalid streamlines using a QuickBundles-based clustering algorithm[77]. The best matching endpoint regions for each resulting cluster were identified by majority voting of the contained streamlines. If multiple clusters were assigned to the same pair of regions, they were merged. Streamlines that were not assigned to any cluster or that fell below a length threshold were labeled as non-connecting.

On the basis of this classification of streamlines, the following metrics were calculated:

1. VC ratio: Number of VCs/total number of streamlines (percentage between 0 and 100).
2. VB: For each bundle that has at least one valid streamline associated with it, this counter is incremented by one (integer number between 0 and 25).
3. IB: With 25 bundles in the ground truth, each having two endpoint regions, there are 1275 possible combinations of endpoint regions. Taking the 25 VBs out of the equation, 1250 potential IBs remain (integer number between 0 and 1250).
4. Overlap: Proportion of the voxels within the volume of a ground truth bundle that is traversed by at least one valid streamline associated with the bundle. This value shows how well the tractography result recovers the original volume of the bundle (percentage between 0 and 100).
5. Overreach: Fraction of voxels outside the volume of a ground truth bundle that is traversed by at least one valid streamline associated with the bundle

over the total number of voxels within the ground truth bundle. This value shows how much the VCs extend beyond the ground truth bundle volume (percentage between 0 and 100). This value is always zero for the traditional definition of a VC, but can be non-zero for the non-stringent criteria we adopted in our study.

Following previously defined criteria of evaluation[79], our study is mainly about accuracy with respect to the reference, rather than reproducibility or robustness of tractography.

**Statistical multi-variable analysis**. Effects of the experimental settings were investigated in a multivariable linear mixed model. The experimental variables describing the methods used for the different submissions were included as fixed effects (Fig. 2b). The VC ratio, the VB count, the IB count, the bundle overlap percentage, and the bundle overreach percentage were modeled as dependent variables, each of which is used for the calculation of a separate model. The submitting group was modeled as a random effect. The software SAS 9.2, Proc Mixed, SAS Institute Inc., Cary, NC, USA, was used for the analysis.

**Data availability**. The authors declare that the data supporting the findings of this study are available within the paper and its Supplementary Information files. The ISMRM 2015 Tractography Challenge data sets and the submitted tractograms are available under doi.org/10.5281/zenodo.572345 and doi.org/10.5281/zenodo.840086, respectively.

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

## Acknowledgements

K.H.M.-H. was supported by the German Research Foundation (DFG), grants MA 6340/10-1, MA 6340/12-1. M.D. was supported by the NSERC Discovery Grant program as well as the institutional Université de Sherbrooke Research Chair in Neuroinformatics. C.M.W.T. was supported by a grant (No. 612.001.104) from the Physical Sciences division of the Netherlands Organization for Scientific Research (NWO). The research of H.Y.M., S.D., S.S., A.M.H., and A.L. was supported by VIDI grant 639.072.411 from NWO. The research of F.G. was funded by the Chinese Scholarship Council (CSC). M.Ch. was supported by the Alexander Graham Bell Canada Graduate Scholarships-Doctoral Program (CGS-D3) from the Natural Sciences and Engineering Research Council of Canada (NSERC). M.C. was supported by the Investigator Award No. 103759/Z/14/Z from the Wellcome Trust. C.C.H. was supported by DFG SFB grants 936/A1, Z3 and TRR 169/A2. The research of J.-P.T., D.R., M.B., A.A., A.L., and A.D. was supported by the Center for Biomedical Imaging (CIBM) of the Geneva-Lausanne Universities and the EPFL, as well as the foundations Leenaards and Louis-Jeantet, and by the Swiss National Science Foundation grants 205321_144529 and 31003A_157063. W.E.R. was supported by CA90246 from National Cancer Institute. The research of Y.F., C.G., Y.W., J.M., H.R., Q.L., and C.-F.W. was supported by grant 61379020 from National Nature Science Foundation of China. C.-F.W. was supported by NIH grants P41EB015902 and P41EB015898.

## Author contributions

K.H.M.-H., M.D., and J.-C.H. performed the data analysis and wrote the paper with input from all authors. P.F.N. and B.S. designed the phantom. P.F.N. and J.-C.H. supported the data analysis and J.-C.H. handled the Tractometer scoring and evaluation metrics proposed. M.-A.C. and E.G. developed the clustering and bundle recognition algorithm for the relaxed scoring system. K.H.M.-H., P.F.N., J.-C.H., E.C., A.D., T.D., B.S., and M.D. coordinated the tractography challenge at the International Society for Magnetic Resonance in Medicine (ISMRM) 2015 Diffusion Study Group meeting. T.H.-L. set up the multivariable statistical model. P.F.N. wrote parts of the Online Methods. L.P. and C.C.H. were mentors in the discussion of the paper and neuroanatomical, as well as neuroscientific context. Submissions were made by the following teams: J.Z. team 1; M.Ch. and C.M.W.T. team 2; F.-C.Y. team 3; Y.-C.L. team 4; Q.J. team 5; D.Q.C. team 6; Y.F., C.G., Y.W., J.M., H.R., Q.L., and C.-F.W. team 7; S.D.-G., J.O.O. G., M.P., S.S.-J., and G.G. team 8; S.S.-J., F.R., and J.S. team 9; C.M.W.T., F.G., H.Y.M., S. D., M.F., A.M.H., and A.L. team 10; S.S.-J., G.G., and F.R. team 11; J.O.O.G., M.P., G.G., and F.R. team 12; A.B., B.P., C.B., M.D., S.B., and J.D. team 13; A.S., R.V., A.C., A.Q., and J.Y. team 14; A.R.K., W.H., and S.A. team 15; D.R., M.B., A.A., O.E., A.L., and J.-P.T. team 16; D.R., M.B., A.A., O.E., A.L., and J.-P.T. team 17; H.E.C., B.L.O., B.M., and M.S. N. team 18; F.P., G.P., J.E.V.-R., J.G., and P.M.T. team 19; F.D.S.R., P.L.L., L.M.L., R.B., and F.D.'A. team 20.

## Additional information

**Competing interests:** The authors declare no competing financial interests.

Klaus H. Maier-Hein[1], Peter F. Neher[1], Jean-Christophe Houde[2], Marc-Alexandre Côté[2], Eleftherios Garyfallidis[2,3], Jidan Zhong[4], Maxime Chamberland[2], Fang-Cheng Yeh[5], Ying-Chia Lin[6], Qing Ji[7], Wilburn E. Reddick[7], John O. Glass[7], David Qixiang Chen[8], Yuanjing Feng[9], Chengfeng Gao[9], Ye Wu[9], Jieyan Ma[10], Renjie He[10], Qiang Li[10,11], Carl-Fredrik Westin[12], Samuel Deslauriers-Gauthier[2], J.Omar Ocegueda González[13], Michael Paquette[2], Samuel St-Jean[2], Gabriel Girard[2], François Rheault[2], Jasmeen Sidhu[2], Chantal M.W. Tax[14,15], Fenghua Guo[14], Hamed Y. Mesri[14], Szabolcs Dávid[14], Martijn Froeling[16], Anneriet M. Heemskerk[14], Alexander Leemans[14], Arnaud Boré[17], Basile Pinsard[17,18], Christophe Bedetti[17,19], Matthieu Desrosiers[17], Simona Brambati[17], Julien Doyon[17], Alessia Sarica[20], Roberta Vasta[20], Antonio Cerasa[20], Aldo Quattrone[20,21], Jason Yeatman[22], Ali R. Khan[23], Wes Hodges[24], Simon Alexander[24], David Romascano[25], Muhamed Barakovic[25], Anna Auría[25], Oscar Esteban[26], Alia Lemkaddem[25], Jean-Philippe Thiran[25,27], H.Ertan Cetingul[28], Benjamin L. Odry[28], Boris Mailhe[28], Mariappan S. Nadar[28], Fabrizio Pizzagalli[29], Gautam Prasad[29], Julio E. Villalon-Reina[29], Justin Galvis[29], Paul M. Thompson[29], Francisco De Santiago Requejo[30], Pedro Luque Laguna[30], Luis Miguel Lacerda[30], Rachel Barrett[30], Flavio Dell'Acqua[30], Marco Catani[30], Laurent Petit[31], Emmanuel Caruyer[32], Alessandro Daducci[25,27], Tim B. Dyrby[33,34], Tim Holland-Letz[35], Claus C. Hilgetag[36], Bram Stieltjes[37] & Maxime Descoteaux[2]

[1]Division of Medical Image Computing, German Cancer Research Center (DKFZ), Heidelberg, 69120, Germany. [2]Sherbrooke Connectivity Imaging Lab (SCIL), Université de Sherbrooke, Sherbrooke, QC J1K 0A5 QC, Canada. [3]Department of Intelligent Systems Engineering, School of Informatics and Computing, Indiana University, Bloomington, IN 47408, USA. [4]Krembil Research Institute, University Health Network, Toronto, Canada M5G 2C4. [5]Department of Neurological Surgery, University of Pittsburgh School of Medicine, Pittsburgh, PA 15213, USA. [6]IMT—Institute for Advanced Studies, Lucca, 55100, Italy. [7]Department of Diagnostic Imaging, St. Jude Children's Research Hospital, Memphis, TN 38105, USA. [8]University of Toronto Institute of Medical Science, Toronto, Canada M5S 1A8. [9]Institute of Information Processing and Automation, Zhejiang University of Technology, Hangzhou, 310023 Zhejiang, China. [10]United Imaging Healthcare Co., Shanghai, 201807, China. [11]Shanghai Advanced Research Institute, Shanghai, 201210, China. [12]Laboratory of Mathematics in Imaging, Harvard Medical School, Boston, MA 02215, USA. [13]Center for Research in Mathematics, Guanajuato, 36023, Mexico. [14]PROVIDI Lab, Image Sciences Institute, University Medical Center Utrecht, Utrecht, 3508, The Netherlands. [15]Cardiff University Brain Research Imaging Centre, School of Psychology, Cardiff University, Maindy Road, Cardiff, CF24 4HQ, UK. [16]Department of Radiology, University Medical Center Utrecht, Utrecht, 3508, The Netherlands. [17]Centre de recherche institut universitaire de geriatrie de Montreal (CRIUGM), Université de Montréal, Montreal, QC, Canada H3W 1W5. [18]Sorbonne Universités, UPMC Univ Paris 06, CNRS, INSERM, Laboratoire d'Imagerie Biomédicale (LIB), 75013 Paris, France. [19]Center for Advanced Research in Sleep Medicine, Hôpital du Sacré-Coeur de Montréal, Montreal, Canada H4J 1C5. [20]Neuroimaging Unit, Institute of Bioimaging and Molecular Physiology (IBFM), National Research Council (CNR), Policlinico Magna Graecia, Germaneto, 88100 CZ, Italy. [21]Institute of Neurology, University Magna Graecia, Germaneto, 88100 CZ, Italy. [22]Institute for Learning & Brain Sciences and Department of Speech & Hearing Sciences, University of Washington, Seattle, WA 98195, USA. [23]Departments of Medical Biophysics & Medical Imaging, Schulich School of Medicine and Dentistry, Western University, 1151 Richmond St N, London, ON, Canada N6A 5C1. [24]Synaptive Medical Inc., MaRS Discovery District, 101 College Street, Suite 200, Toronto, ON, Canada M5V 3B1. [25]Signal Processing Lab (LTS5), Ecole Polytechnique Federale de Lausanne, Lausanne, 1015, Switzerland. [26]Biomedical Image Technologies (BIT), ETSI Telecom., U. Politécnica de Madrid and CIBER-BBN, Madrid, 28040, Spain. [27]Department of Radiology, University Hospital Center (CHUV) and University of Lausanne (UNIL), Lausanne, 1011, Switzerland. [28]Medical Imaging Technologies, Siemens Healthcare, Princeton, NJ 08540, USA. [29]Imaging Genetics Center, Stevens Neuro Imaging and Informatics Institute, Keck School of Medicine of USC, Marina del Rey, CA 90033, USA. [30]NatBrainLab, Institute of Psychiatry, Psychology & Neuroscience, King's College London, London, SE5 8AF, UK. [31]Groupe d'imagerie Neurofonctionnelle—Institut des Maladies Neurodégénératives (GIN-IMN), UMR5293 CNRS, CEA, University of Bordeaux, Bordeaux, 33000, France. [32]Centre national de la recherche scientifique (CNRS), Institute for Research in IT and Random Systems (IRISA), UMR 6074 VISAGES Project-Team, Rennes, 35042, France. [33]Danish Research Centre for Magnetic Resonance, Center for Functional and Diagnostic Imaging and Research, Copenhagen University Hospital Hvidovre, Hvidovre, 2650, Denmark. [34]Department of Applied Mathematics and Computer Science, Technical University of Denmark, Kongens Lyngby, 2800, Denmark. [35]Division of Biostatistics, German Cancer Research Center (DKFZ), Heidelberg, 69120, Germany. [36]Department of Computational Neuroscience, University Medical Center Eppendorf, Hamburg, 20246, Germany. [37]University Hospital Basel, Radiology & Nuclear Medicine Clinic, Basel, 4031, Switzerland

