## [Peer Review File · Nature Communications]

Reviewers' comments:

Reviewer #1 (Remarks to the Author):

This paper results from the ISMRM 2015 Challenge aiming at evaluating the robustness of diffusion-based tractography pipelines to infer long white matter bundles of an numerical human brain phantom, and involving 20 different research groups, thus resulting in a large list of co-authors.

The major claim from the authors is that the connectograms built from the proposed pipelines more positives than true white matter bundles, as established using the Tractometer tool.

While providing quantitative estimates of the ratio of false positives, this result cannot be considered as a novel or breakthrough idea. Several publications have already pointed the limitation of diffusion tractography. The authors are invited to read Duffau H, World Neurosurgery, 2014 Jan ; 81(1)-56-8 « The dangers of magnetic resonance imaging diffusion tensor tractography in brain surgery ».

The actual limitations of tractography rely on the fact that most techniques were developed with too few considerations about the underlying neuroanatomy, because till recently, the community was mainly driven by image analysis, MR physics and signal processing experts, with two few connections to the community of neuroanatomists.

In particular, the increased popularity of connectomics approaches originally introduced by Hagman & Sporn and used by most neuroscientists like a black box is today facing the huge variability of obtained results, making it difficult to build the holy grail, eg the human brain structural connectome. The connectivity between cortical areas involves sub-cortical white matter bundles and is highly dependent on the ability with actual data limited in resolution and fiber tracking methods employing too few anatomical priors to infer them accurately. Due to the limited resolution of DW-MRI data with respect to the dimensions of axons and fascicles (even with the Connectome gradient machines), to the limitations of local angular models of the diffusion process, and to the limitations of fiber tracking methods, claiming that it is a challenging task is obvious and cannot be considered as a novel message to the community. The illness-posed nature of the problem was already pointed out by the diffusion community back in the 2000's.

The methods presented in the paper can be also be discussed. The human brain phantom was designed from the global tractography approach of Reseirt et al making use of a generative approach to give birth to, move and connect spins and to create fibers, followed by a manual selection of fibers by a neuroanatomist to create long white matter bundles, and finally used to recompose a DW-MRI dataset using a linear combination of sticks, ball and isotropic compartments and tuned using the original DW image. This approach can be discussed, because the construction of the phantom rely on models and methods that are also the target feature to be evaluated, thus making the evaluation circular. Even more, the HCP dataset used to tune the signal of the output DW-MRI dataset remains far from being representative of a dense sampling of the q-space in order to used most advances local

diffusion models.

In addition, the fact that no competitor used the most recent local models such as propagator imaging or the most advanced tractography approaches like global approaches informed with anatomical priors makes the relevance of the conclusion limited as it points out their already well-known limitations.

In conclusion, while providing a detailed summary of the ISMRM 2015 competition, this paper does not contribute to the emergence of novelty in the field of diffusion MRI and connectomics. This paper should rather be submitted to journals focused on methods and cannot be published in Nature Communications.

Reviewer #2 (Remarks to the Author):

Summary

The paper by Maier-Hein et al. compiles all the submissions and results of an open competition that aimed to quantify errors made by diffusion MRI tractography algorithms. Both the organisers and all participants (who also co-authored the paper) have done a thorough job for which they must be warmly congratulated.

The paper addresses the issue of quantifying errors in tractography - the now mainstream set of techniques for mapping brain connections in vivo. This is a challenging problem due to the lack of ground truth on human brain connections. The approach that has been taken here was to artificially generate ground-truth connections and data using simulations. This is an approach that has been taken before by numerous investigators and algorithm developers. The novelty in this paper is that (i) the simulations attempt to mimic the complexities of real brain geometries, and (ii) by using an open competition framework, the authors were able to compare an exhaustive list of tractography algorithms covering most of the available methods, as well as evaluate the impact of various choices of data processing pipelines.

The main conclusion of the paper is unambiguously spelt out in the title: "Tractography-based connectomes are dominated by false-positive connections". This is a very strong statement, and I would like to unambiguously say that the material presented in this paper does not support such a strong statement. I urge the authors to tone down their claims in light of my comments below. This is an important paper, endorsed (co-authored) by a large number of key figures in the field, and it is likely to be impactful.

It is worth re-emphasising the purpose of this paper. The authors acknowledge early on that the caveats of diffusion MRI tractography algorithms are well known and well documented. The purpose of this paper is to actually quantify the effect that these caveats have on estimating connectivity. I believe that the authors used an interesting approach, but I don't think that they can claim to have quantified the amount of errors in tractography, and therefore the statement made in the title and other parts of the paper is not justified in my opinion.

General comments on the approach

In order to explain my view, I first need to recap the approach taken in this paper in brief and basic terms. In essence, the authors of this paper ask the following simple question: given a continuous 3D vector field, can we recover the field after discretising it? The actual sequence of steps that are followed is: (i) create a continuous vector field, (ii) discretise the vector field to get a range of orientations per bin/voxel, thereby losing some spatial specificity (iii) create artificial diffusion data based on the discrete orientations, and (iv) add "acquisition artefacts".

Now, with sufficient data of sufficient quality, steps (iii) and (iv) can be undone, given that these are simulated data. In fact, the authors even included some results where these two steps are not applied to the vector field at all in order to rule out effects due to these last two steps. This was a particularly important thing to have done because the final data used in the competition was of very low quality indeed. I will come back to this point a little later.

So essentially, it is all about step (ii): how much of the vector field can be recovered after it has been spatially discretised? Before we get to the details of how the authors quantify this, it is worth contemplating the implications of this question in terms of the overall stated aim of the paper (which is to quantify errors in tractography). Firstly, the amount of error (between recovered and original field) will depend on the geometry of the field: simpler fields (e.g. straight lines) are easier to recover from discrete versions. It also depends on the resolution of the discrete grid: the higher the resolution of the grid the better the recovered field. This is a simple point, but amazingly the authors have only used a single grid resolution (2mm) and yet they make quantitative claims about tractography in general. The authors claim that their results are independent of data quality (by which they also mean spatial resolution) because one of their results includes using the "ground truth" orientations, but these have also been discretised, i.e. step (ii) has been applied (but not steps iii and iv). Evidently, had they used the original vector field (i.e. very high resolution), they would have recovered it in its entirety with zero error.

It follows that the claims of the paper are not general but dependent on the geometric and topological properties of the true continuous orientation field as well as on the resolution of the grid. Any quantitative measure of error therefore is linked to these factors and cannot be trivially generalised. Anyone can design continuous orientation fields that challenge tractography to various degrees depending on how they are affected by discretisation. Error will change accordingly.

Quantification and why it is biased

The approach taken by the authors to quantify errors in tractography is worth examining in detail since it is the basis for the main claim: that tractography is dominated by false positives.

Participants in the contest were asked to submit a set of candidate streamlines (continuous paths through the discrete orientation field). These are then compared to the simulated

bundles of streamlines and classified as valid or invalid. Interestingly, the authors focus on the occurrences of valid vs invalid bundles (groups of streamlines) rather than the occurrences of valid vs invalid streamlines. They find that invalid bundles outnumber valid ones by a factor of 4, which is the basis of their main claim.

This is problematic for two reasons. First, invalid streamlines are grouped into so-called invalid bundles with a clustering algorithm, which ultimately determines their number. Unless these invalid bundles have the same properties as valid ones in terms of their shapes/lengths/widths/etc., there is no reason to expect them to have the same clustering properties. In fact, they are more likely to contain multiple clusters. Secondly, invalid bundles connect regions that the authors simulated as non-connected. There are 25 simulated connections (25 pairs of connected regions), and thus 275 possible wrong-connections, outnumbering correct connections by a factor of 11. The authors report absolute numbers of valid/invalid bundles, but given the above, they should be reporting relative numbers, in which case their conclusions will change dramatically.

Interestingly, hidden deep into the supplementary material is the fact that by calculating the number of correct streamlines (as opposed to bundles), at least one of the entries scored a whopping 92% correct and 8% incorrect. That is very far from what the authors conclude when they group results into bundles as they have done for their main results. In essence, what they have done is they have grouped the 8% incorrect streamlines in such a way that these 8% are clustered into a sufficiently large number of bundles that they now outnumber correct bundles by a factor of 4!

There are two other points to be made on the question of quantification. First, the major analyses are done as binary valid/invalid, whereas it is conceivable that some of the reconstructed connections might follow a correct route for a while before deviating from the correct trajectory. This is somewhat captured by their measure of overlap vs overreach, but is only considered in the cases of "valid bundles". Secondly, there is no accounting for uncertainty associated with streamlines. There is a host of techniques for tractography that incorporate uncertainty, but the scoring system seems to ignore this. Although the authors claim that some of the submissions included probabilistic tractography methods, I don't see how their scoring system accounts for the uncertainty information associated with these methods, particularly once streamlines are grouped into bundles that are scored as a whole.

MRI data

An important aspect of this paper is the very low quality of the data that have been shared with the contestants. In particular, the low number of directions (30) and relatively low b-value (1000 s/mm²) means that the deconvolution involved in undoing step (iii) of the simulation chain is particularly hard. While the authors claim that this choice was motivated by the fact that most data out there are of similarly low quality, the conclusions made in this paper (and the title itself) sound universal, and independent of data quality. Another argument made by the authors is that spatial resolution is not a factor. I think it is an incredibly important factor: even in their ground-truth tractography where they have removed steps (iii) and (iv) of the processing, the authors still discretised the field. Their

results therefore are dependent on the spatial resolution that they have chosen to use. Nowadays, MR technology enables $\sim 1\text{mm}$ spatial resolution in vivo. It would be straightforward for the authors to simulate such data and assess the benefits in terms of reducing errors on particularly tricky brain connections.

Don't get me wrong

The arguments that I have put forward in this review may give the false impression that my view on tractography is filtered by pink goggles. It is not. I do think that diffusion tractography can be error prone, but I also think that quantification of this error is very hard indeed. The authors focused on a very specific source of errors with tractography; I would say that it is arguably less important than other much more fundamental problems.

Conclusion

This paper presents an interesting and valuable approach to the question of validating tractography. In addition, the general approach as well as the current framework and data will be an invaluable resource for the community to develop better methods in the future. However, I feel that the negative tone of this paper is both unjustified and potentially extremely harmful to the field. This paper does not quantify the occurrence of false positives in tractography, and therefore it cannot claim that tractography-based connectomes are dominated by false positives. At most, this paper shows that there are situations where false positives are likely to occur (to an unknown extent) due to lack of spatial resolution.

Written by: Saad Jbabdi

Reviewer #3 (Remarks to the Author):

In their paper entitled 'Tractography-based connectomes are dominated by false-positive connections' Maier-Hein et al. analysed the reproducibility of tractography across 20 research groups. While the method is ingenious, my enthusiasm has been tempered by the lack of new findings.

In general:

Indeed, it is quite difficult to disprove findings obtained with tractography due to the lack of a gold standard model with regard to white matter anatomy in humans. While post-mortem validation studies do exist, they are sparse and technically difficult to achieve. Therefore, reproducibility of tractography findings remains an important factor.

However, here the authors circumvent the problems associated with the lack of such a model by ingeniously building a mock white matter gold standard based on the best tractography dissections available - as far as they know. They called their gold standard a 'synthetic ground truth' and subsequently derived a diffusion imaging dataset from this material. This dataset has been sent to 20 research groups for preprocessing and dissections in order to assess the reproducibility of the findings. Their results indicate a frighteningly low reproducibility across research groups, with 4 times more invalid bundles

compared to the 'synthetic ground truth'.

Besides the ingenious approach described above, it is difficult to extract an important and productive scientific message from this paper. As it reads, the paper is focused on tractography caveats and does not offer a viable improved approach nor solution. Such a report may have a negative impact on the field of tractography by putting forth negative and undue influence on the use of methods available to researchers of white matter anatomy and function without providing any viable alternatives or solutions.

There is a lack of novelty in the findings. The limitations of tractography have previously been demonstrated in many publications and books, particularly with regard to the organisation of complex fibres (such as Catani 2007, Jones 2008).

Additionally, the findings reported are not associated solely to the use of tractography but also to magnetic resonance imaging methods in general.

Tractography may have errors, but so does MR based cortical thickness and voxel based morphometry (Zilles et al. 2015) functional neuroimaging (Logthetis 2008) and T1 based myelin quantification (Sandrone et al. in press), voxel based lesion symptom mapping (Mah et al. Brain 2015). All of these approaches are limited because they assess the features of the living human brain based on an indirect magnetic resonance approach. Indirect measures are not exempt of errors.

More specifically:

The authors did not account for operator dependant errors, therefore shifting all the blame on tractography. This is indeed a limitation in their methods that should at least be mentioned.

The text indicates "that Some of these false-positive bundles resemble previously reported pathways identified by in-vivo tractography, such as the frontal aslant tractor the vertical occipital fasciculus" and later states "The existence of the FAT, SFOF and VOF is controversial (41,42,49,54)". Do the authors suggest that these tracts do not exist in humans? This statement requires clarification because it constitutes a direct attack on previous work, being mindful that the FAT and SFOF have been validated with post-mortem dissection. Disproving these findings with a 'synthetic ground truth' will challenge the credibility of the rest of their findings.

Reviewers' comments:

Reviewer #1 (Remarks to the Author):

This paper results from the ISMRM 2015 Challenge aiming at evaluating the robustness of diffusion-based tractography pipelines to infer long white matter bundles of an numerical human brain phantom, and involving 20 different research groups, thus resulting in a large list of co-authors.

R1.1: We thank the Reviewer for this brief summary. We would like to clarify one important point right in the beginning, since it has an impact on many of the following points raised by the Reviewer. Our study does not primarily aim at evaluating robustness. To briefly review the most important "criteria of validation", as given by Jannin et al. (CARS 2002):

Accuracy: Degree to which a measurement is true or correct. Difference between computed values and theoretical values.

Reproducibility or Reliability: Intrinsic to the process, regards resolution at which process is repeatable.

Robustness: Performance in the presence of disruptive factors such as intrinsic data variability, pathology, or inter-individual anatomic or physiologic variability.

In line with these definitions, our study is mainly about accuracy with respect to the reference. This was now clarified in the Online Methods, sec. "Performance metrics and evaluation".

The major claim from the authors is that the connectograms built from the proposed pipelines more false positives than true white matter bundles, as established using the Tractometer tool.

While providing quantitative estimates of the ratio of false positives, this result cannot be considered as a novel or breakthrough idea. Several publications have already pointed the limitation of diffusion tractography. The authors are invited to read Duffau H, World Neurosurgery, 2014 Jan ; 81(1)-56-8 « The dangers of magnetic resonance imaging diffusion tensor tractography in brain surgery ».

R1.2: We agree that we are not the first to discuss limitations of tractography. Consequently, we stated in the beginning of the Introduction: "the advantages and shortcomings of tractography have been widely debated^{1,5-10}". The novelty of our work lies in the validation method, the extensive challenge experiments providing a spectrum of experimental setups and the related quantitative findings pointing at future general directions for methodological development.

Regarding Duffau et al. (2014), it is important to note that the article is an opinion article about whether or not tractography should be used in the context of brain surgery planning. It mainly discusses findings from Feigl et al. (2014), who compare different tractography software packages. The comparison was performed using images with only 12 diffusion directions and a large slice thickness of 5mm. Due to the lack of quantitative alternatives, the comparison was based on the visual rating of screenshots by three examiners. Mean visual grade for "anatomic accuracy" was 2.2 ± 0.6 and "incorrectly displayed fibers" were rated with 2.5 ± 0.6 . The authors especially focus on their finding that the grades between the best and the lowest ranking package differed statistically significantly. They conclude that tractography requires further validation before being applied for surgery planning.

In comparison, the present study is more advanced in terms of methodology compared to the papers by Feigl et al. and Duffau et al. However, the articles provides a very good motivation and preparation for our work, which is why we included these references in the revised version of our introduction (paragraph 1).

The actual limitations of tractography rely on the fact that most techniques were developed with too few considerations about the underlying neuroanatomy, because till recently, the community was mainly driven by image analysis, MR physics and signal processing experts, with too few connections to the community of neuroanatomists.

R1.3: The need for close collaboration between these fields is undebatable and also recognized by us. There are several things that we would like to respond to this statement and that we have also taken into consideration when revising the 4th and the last paragraph of our discussion:

First, even in the hypothetical situation where we assumed that we had perfect knowledge of brain anatomy somehow, it is still currently unclear to what degree this information could be leveraged in tractography when reconstructing *personalized* tractograms.

Second, there is certainly some unexplored potential in anatomical priors that we also discuss in our study and that we also intensely discussed with some of the most renowned neuroanatomists in the field that are coauthoring our study. However, we must not forget that our current knowledge of brain anatomy also comes with its own set of limitations, and that it is currently not known how to optimally model and leverage such information for tractography.

Third, as the Reviewer points out, there exist strong opinions about current tractography limitations and their origins. We strongly believe that experimental analysis and validation will be the only way to catalyze this discussion and find a way out of the potential local minimum that the field might be stuck in. Our study may be an important step in this direction: It reveals fundamental issues with the current problem formulation in tractography and at the same time provides a way to quantify the next generation of tractography algorithms proposed.

In particular, the increased popularity of connectomics approaches originally introduced by Hagman & Sporn and used by most neuroscientists like a black box is today facing the huge variability of obtained results, making it difficult to build the holy grail, eg the human brain structural connectome.

R1.4: Indeed, our findings regarding tractography are particularly important for the field of connectomics, where it is currently not possible to check for the correctness of all the suggested connections in the structural connectivity matrix. Thus, we expanded the discussion of these important aspects in the manuscript (see Discussion, paragraph 5).

Our results showed that the variability of output obtained by different “black boxes” is certainly one, but not the only problem that tractography is facing today. Tractography has important principal limitations, even when choosing the best black boxes currently available. However, it is also important to note that tractography and connectomics in its current form is *not useless*: For example, certain graph indices might represent highly useful early indicators of a psychiatric disease. But still, we have to be aware of what aspects of tractography are reliable and what aspects are not.

The connectivity between cortical areas involves sub-cortical white matter bundles and is highly dependent on the ability with actual data limited in resolution and fiber tracking methods employing too few anatomical priors to infer them accurately.

R1.5: We agree that there exist bundles – such as sub-cortical white matter bundles – that are even more difficult to obtain with tractography than some of the well-known larger bundles included in our phantom. We discuss possible extensions of the phantom in the 2nd and 5th paragraph of the discussion.

We also agree that the quality at which we are able to resolve bundles relates to the resolution of the images used and also on the anatomical priors employed (cf. Dyrby et al., Neuroimage 2015). We

added additional experiments that incorporate existing anatomical priors as well as high resolution images (Supplementary Figs. 6 and 7). The results supported the findings that we present in our study.

Due to the limited resolution of DW-MRI data with respect to the dimensions of axons and fascicles (even with the Connectome gradient machines), to the limitations of local angular models of the diffusion process, and to the limitations of fiber tracking methods, claiming that it is a challenging task is obvious and cannot be considered as a novel message to the community. The illness-posed nature of the problem was already pointed out by the diffusion community back in the 2000's.

R1.6: We believe that our findings can trigger a new wave of innovation in tractography, especially when published in a leading journal like Nature Communications. One unique aspect of this study is the independent distribution of methods used (cf. Reviewer 2: “an exhaustive list of tractography algorithms covering most of the available methods”), demonstrating for the first time that it is not just a matter of fine tuning methods to achieve accurate tractography. We also show now that higher angular or spatial resolution and higher b-values will not suffice in solving the issues. We identify crucial aspects in the processing pipelines that contribute most strongly to accuracy vs error.

The results will help scientists interpreting their tractography findings and provide an important message that will bring forward our field. The previous work in the context of tractography validation is cited and discussed in the manuscript (especially in the Introduction 1st and 2nd paragraphs), hopefully convincing the Reviewer that our findings are not obvious and bear a truly novel message to the community. A positive indication that our findings are appreciated by the community is the considerable interest that the preprints of our manuscript received on biorxiv (reads 6000, downloads 2000, altmetric score 130).

The methods presented in the paper can be also be discussed. The human brain phantom was designed from the global tractography approach of Reese et al making use of a generative approach to give birth to, move and connect spins and to create fibers, followed by a manual selection of fibers by a neuroanatomist to create long white matter bundles, and finally used to recompose a DW-MRI dataset using a linear combination of sticks, ball and isotropic compartments and tuned using the original DW image. This approach can be discussed, because the construction of the phantom rely on models and methods that are also the target feature to be evaluated, thus making the evaluation circular.

R1.7: This – important – issue of circular evaluation and its potential impact on the findings is indeed discussed at length in the 2nd paragraph of the Discussion (“self-validation issue”). Being aware of this pitfall, we were very cautious in not making claims that could be an artifact of this circularity.

Even more, the HCP dataset used to tune the signal of the output DW-MRI dataset remains far from being representative of a dense sampling of the q-space in order to used most advances local diffusion models.

R1.8: We acknowledge the fact that it is hard to generalize findings from a given set of diffusion models to all existing models. This is why we added tractography experiments that were performed directly on the gold standard directions, thus simulating a hypothetical model that is able to remove all the noise from the signal and then to correctly and accurately estimate all the different diffusion directions existent in the given voxel's ground truth. This procedure enabled us to establish an “upper bound” of the influence of more powerful diffusion models. In the revised version of the manuscript we further extended these experiments by performing tractography on the ground truth peaks at multiple resolutions going down to 0.5 mm isotropic resolution (Supplementary Fig. 6, Supplementary Notes 3, see also comments from and responses to Reviewer 2). We show that

tractography is ill-posed even when applying such a hypothetical “optimal” model at high spatial resolution, which by far outperforms the most advanced local diffusion models.

In the revised version of the manuscript (Discussion, 1st paragraph), we now explicitly mention the local modeling in light of these findings: “[...] show how the computational problem of tractography goes far beyond the local modeling and reconstruction of fiber directions^{1,8} and issues of data quality.”

In addition, the fact that no competitor used the most recent local models such as propagator imaging or the most advanced tractography approaches like global approaches informed with anatomical priors makes the relevance of the conclusion limited as it points out their already well-known limitations.

R1.9: In our view, as we argued above, we do see a number of novel insights that go well beyond the well-known limitations of tractography. The Reviewer suggests that there exist more advanced approaches that may solve the problem that we report. For the advanced local modeling techniques, our results that we provided in response to the previous comment show that they will not be able to bridge the gap between directionality and connectivity. For the recently proposed global tractography approaches, we provide new experimental results that analyze the performance of currently available global and anatomically informed methods.

In the following sections, we describe the novel experiments step by step from step (I) to step (III). The experiments were also included in the revised version of the manuscript (Supplementary Notes 3 and Supplementary Fig. 7).

In step (I), we identified all tractography methods that combine a global approach with anatomical constraints. The following approaches have been identified:

[1] Christiaens et al. “Atlas-Guided Global Tractography: Imposing a Prior on the Local Track Orientation”, MICCAI Workshop on Computational Diffusion MRI, 2014.

This paper introduces a local prior on the tract orientation. However, an improved reconstruction of the underlying peak directions for tractography will not solve the problem as our experiments on the ground truth peaks show. The method is not openly available.

[2] Lemkaddem et al. “Global Tractography with Embedded Anatomical Priors for Quantitative Connectivity Analysis”, Frontiers in Neurology, 2014.

This approach specializes on connectivity analysis and thus seeks for reconstructing fibers that terminate in the gray matter. The approach requires an initial library of fiber tracts that is obtained using classical tractography (!). The classical tractography is augmented by the following steps: Prematurely ending fibers are then either extended into the gray matter or deleted if too far from the gray matter. Fibers are then reweighted regarding to their estimated contribution to the measured MRI signal using MCMC-based optimization. Neither extension nor deletion of fibers will solve the issues discussed in our manuscript. Also, the reweighting procedure does not help as already shown and discussed in the previous version of our manuscript. The method is not openly available.

[3] Christiaens et al. “Imposing label priors in global tractography can resolve crossing fiber ambiguities”, Annual meeting ISMRM, 2015.

This is the only method that approaches the issue of invalid connections by using a white matter atlas as prior knowledge. The method is not openly available, but we are in contact with the authors and regarding the availability of their method the authors stated:

“The short answer is no, they are not available as ready to use tools. Both methods live in git branches on my system, but I haven't kept them up to date with the main implementation of `tckglobal` so they're not ready to be directly plugged in. I have not further pursued the local track orientation prior because its effect is relatively small. The global label prior, on the other hand, is IMHO very promising, but hinges on having good atlases available. Unfortunately, as far as I know the most complete fibre bundle atlas to date is still based on DTI (Catani). Flavio Dell'Acqua has been working on a CSD-based successor for years, but so far the results have not been published. I should work more on the label priors at some point, but in the absence of a good atlas I have not been very motivated...”

Regarding the inclusion of their method in our analysis the authors further stated:

“I think you might just say that these studies are a proof of concept, rather than a ready to use method?”

In step (II), due to the mentioned issues with the approaches that combine global optimization with anatomical constraints, we decided to perform experiments that separately analyze the two aspects. Thus, we identified openly available approaches that either perform global optimization or that use anatomical constraints. We chose the following three approaches:

[4] Smith et al. “Anatomically-constrained tractography: improved diffusion MRI streamlines tractography through effective use of anatomical information”; MRtrix

This approach is openly available and incorporates anatomical constraints based on deterministic or probabilistic streamline tractography (ACT). In Supplementary Fig. 7, this method is referred to as “ACT Deterministic” and “ACT Probabilistic”.

[5] Reisert et al. “Global fiber reconstruction becomes practical”, MITK

This approach implements global tractography without anatomical priors. In Supplementary Fig. 7, this method is referred to as “MITK Global”.

[6] Christiaens et al. “Global tractography of multi-shell diffusion-weighted imaging data using a multi-tissue model”, MRtrix

This approach as well implements global tractography without anatomical priors. In Supplementary Fig. 7, this method is referred to as “MRtrix Global”.

In step (III), we incorporated two newly implemented tractography approaches that combine the above cited global tractography methods with posteriorly applied anatomical constraints. This was achieved on basis of the constraints described by Smith et al. [4], thus excluding all fibers that do not start and end in the gray matter (brain stem manually added to gray matter) are removed, excluding all fibers that pass CSF and excluding all fibers with more than 25% of their length outside of the white matter. In Supplementary Fig. 7, the newly implemented methods are referred to as “MITK Global ACT” and “MRtrix Global ACT”.

We performed the series of experiments using the following methodology: First, the diffusion-weighted image was denoised and corrected for headmotion and distortions using MRtrix (`dwidenoise` & `dwipreproc`, <http://www.mrtrix.org/>). Multi-tissue response functions were estimated

using MRtrix (*dwi2response*). Single shell spherical deconvolution was performed using the white matter response function. A map of white matter, gray matter and CSF was estimated from the T1 image using the MRtrix command *5ttgen*. Deterministic and probabilistic (*iFOD2*) ACT [4] was performed using MRtrix with 100000 seeds randomly placed inside the brain mask. A minimum fiber length threshold of 20mm was applied. MRtrix global tractography [6] was performed using a 2-tissue WM/CSF-model within a WM mask, as recommended by the MRtrix team. The white matter mask was estimated from the T1 image. The particle weight was set to 0.05 and the number of iterations to 1e9. MITK global tractography [5] was performed on a Q-Ball reconstruction of the diffusion-weighted image within the whole brain. The particle weight was set to 0.0007, the particle width to 0.5 and the number of iterations to 1e9.

Supplementary Fig. 7. Results of the new experiments (colored) using different combinations of global tractography and anatomically constrained tractography (ACT) in conjunction with the scores of the original submissions (gray).

The results strongly support our previous claims, showing that the newly incorporated methods perform similar to the other approaches and that they do not solve the issues discussed in the manuscript (see Supplementary Fig. 7).

In conclusion, while providing a detailed summary of the ISMRM 2015 competition, this paper does not contribute to the emergence of novelty in the field of diffusion MRI and connectomics. This paper should rather be submitted to journals focused on methods and cannot be published in Nature Communications.

R1.10: The substantial concerns in this review helped us to design several additional experiments that were necessary to better support our claims with respect to the fundamental ambiguities inherent to tract reconstruction. We put a lot of effort in reporting and discussing our findings adequately, and we hope that we could convince the Reviewer that the findings will significantly progress and not harm the field of tractography.

Reviewer #2 (Remarks to the Author):

Summary

The paper by Maier-Hein et al. compiles all the submissions and results of an open competition that aimed to quantify errors made by diffusion MRI tractography algorithms. Both the organisers and all participants (who also co-authored the paper) have done a thorough job for which they must be warmly congratulated.

The paper addresses the issue of quantifying errors in tractography - the now mainstream set of techniques for mapping brain connections in vivo. This is a challenging problem due to the lack of ground truth on human brain connections. The approach that has been taken here was to artificially generate ground-truth connections and data using simulations. This is an approach that has been taken before by numerous investigators and algorithm developers. The novelty in this paper is that (i) the simulations attempt to mimic the complexities of real brain geometries, and (ii) by using an open competition framework, the authors were able to compare an exhaustive list of tractography algorithms covering most of the available methods, as well as evaluate the impact of various choices of data processing pipelines.

R2.1: This summary is very much to the point. Just a small note: we also quantify *success*, not only errors made by tractography algorithms. In other words, there are also positive messages coming from the experiments, and we tried to further emphasize them in the revised version of the manuscript. Particularly, contributors find most bundles of the dataset to a larger or smaller extent. We now put more emphasis on the positive messages in the revised version of the manuscript.

The main conclusion of the paper is unambiguously spelt out in the title: “Tractography-based connectomes are dominated by false-positive connections”. This is a very strong statement, and I would like to unambiguously say that the material presented in this paper does not support such a strong statement. I urge the authors to tone down their claims in light of my comments below. This is an important paper, endorsed (co-authored) by a large number of key figures in the field, and it is likely to be impactful.

R2.2: We agree with the Reviewer that this is a strong statement. We respond to the suggestion after having discussed the detailed Reviewer’s comments below.

It is worth re-emphasizing the purpose of this paper. The authors acknowledge early on that the caveats of diffusion MRI tractography algorithms are well known and well documented. The purpose of this paper is to actually quantify the effect that these caveats have on estimating connectivity. I believe that the authors used an interesting approach, but I don’t think that they can claim to have quantified the amount of errors in tractography, and therefore the statement made in the title and other parts of the paper is not justified in my opinion.

R2.3: We certainly agree: Our analysis is not based on real brain connectivity (which is currently hardly accessible as a whole), so our approach cannot quantify anything related to in-vivo connectivity in an absolute sense. We toned down and more precisely qualified statements that could be interpreted wrongly.

An important contribution of our study is the demonstration of how the known and well-documented caveats of tractography behave when going from easy-to-understand artificial situations (as commonly depicted in discussions of tractography caveats) to a more complex scenario that includes the known major bundles in the brain. We show this for a broad spectrum of currently available methods and in several additional experiments at varying image quality.

So yes, we are reporting about the false-positive problem in a quantitative fashion. And we are reporting *trade-offs* between obtaining existing bundles with good overlap and obtaining errors (false positives) in a quantitative fashion. But the conclusions do not rely on the exact numbers obtained from the phantom.

The claim is about the amplitude of effect that we are seeing when introducing only some complexity. The real brain introduces another, even higher level of complexity than our phantom. There is no reason to believe that the observed effects become less relevant in the real brain. Rather, the opposite should be expected: Problems will become even more severe.

General comments on the approach

In order to explain my view, I first need to recap the approach taken in this paper in brief and basic terms. In essence, the authors of this paper ask the following simple question: given a continuous 3D vector field, can we recover the field after discretising it? The actual sequence of steps that are followed is: (i) create a continuous vector field, (ii) discretise the vector field to get a range of orientations per bin/voxel, thereby losing some spatial specificity (iii) create artificial diffusion data based on the discrete orientations, and (iv) add “acquisition artefacts”.

R2.4: This summary was important for us to understand the Reviewer’s view of our approach. Hopefully, we can convince the Reviewer that he missed one very essential point, which is the basis for many of our arguments that we make later to address the Reviewer’s points:

There is an important difference between the recovery of streamlines (connectivity) and the recovery of a continuous vector field (directionality). Our approach validates the recovery of ground truth connectivity (i.e. streamlines). The recovery of a continuous vector field is very different, since it does not include the concept of connectivity. This is why we stated in the introduction:

“Conceptually, the principle of inferring connectivity from local orientation fields can lead to problems as soon as pathways overlap, cross, branch and have complex geometries (Fig. 1)^{8,38,39}.”

It is easy to show that the problem of reconstructing connectivity from a vector field is ill-posed, even if the vector field is given in high spatial resolution or even in a continuous form. In Fig. 1 of the manuscript we conceptually explain this using a simple example (last row of the table). A very common configuration of two tracts leads to many potential hypotheses during tract reconstruction that cannot be resolved, even with the continuous vector field provided.

For the later discussion, it is important to bear in mind that step (i) of our approach we have a model of connectivity on basis of streamlines and not a model of directionality on basis of a continuous vector field.

Now, with sufficient data of sufficient quality, steps (iii) and (iv) can be undone, given that these are simulated data. In fact, the authors even included some results where these two steps are not applied to the vector field at all in order to rule out effects due to these last two steps. This was a particularly important thing to have done because the final data used in the competition was of very low quality indeed. I will come back to this point a little later.

So essentially, it is all about step (ii): how much of the vector field can be recovered after it has been spatially discretised? Before we get to the details of how the authors quantify this, it is worth contemplating the implications of this question in terms of the overall stated aim of the paper (which is to quantify errors in tractography). Firstly, the amount of error (between recovered and original field) will depend on the geometry of the field: simpler fields (e.g. straight lines) are easier to recover from discrete versions. It also depends on the resolution of the discrete grid: the higher the

resolution of the grid the better the recovered field. This is a simple point, but amazingly the authors have only used a single grid resolution (2mm) and yet they make quantitative claims about tractography in general.

R2.5: As argued above, we aim at recovering connectivity and not a continuous vector field, which would be much easier.

We agree with the Reviewer that geometry is important. As the Reviewer mentions, there are examples of simple ground truth connectivity that are easy to recover. But there are also very simple examples that are already ill-posed. This holds even for infinite resolution of the vector field.

One simple example is given in the bottleneck situation in Fig. 1 of the manuscript. While this is a “simple” case, our phantom has “intermediate” complexity. And the real brain is again much more complex.

We also agree with the Reviewer that resolution is relevant. In the design of the challenge, it was important to us to open this challenge to as many teams as possible. One consequence of this approach is that we needed a dataset that everybody can cope with. In consequence, we chose a resolution and acquisition settings representing a typical dMRI dataset that anybody could handle with their pipelines.

In the previous version of the manuscript, higher resolution experiments at 1.25mm were only performed in our in-vivo experiment (Fig. 5 in the manuscript). Here, of course, we could not quantify errors, but we were able to show similar geometrical shapes and features of reconstructed bundles as seen in the phantom dataset. It is most likely fair to assume that **not all** these reconstructed bundles are valid connections.

Additional experiments with higher resolution grids in the phantom were not considered in the previous version of the manuscript. In the revised version of the manuscript, we added a series of experiments where we did not only “undo” steps (iv) and (iii), but where we **additionally** included resolutions down to 0.5 mm isotropic (step ii). In short, the findings did not invalidate any of the conclusions (Supplementary Notes 3 and Supplementary Fig. 6).

In order to assess metric changes with changing resolution, we repeated the deterministic streamline experiment performed on the 2mm isotropic ground truth vector field with the following image resolutions: 2mm, 1.75mm, 1.5mm, 1.25mm, 1mm, 0.75mm and 0.5mm. We used two independent implementations of deterministic peak based streamline tractography for these experiments (GT₁ and GT₂).

Our results show that increasing the resolution improves overlap and overreach scores, but it **does not improve** the scores on invalid bundles (Supplementary Fig. 6). In fact, depending on the method, the number of invalid bundles is even increasing with higher image resolutions (see Supplementary Fig. 6, GT₁). As expected the results confirm that our findings are not an artifact of low resolution, and that the problems persist even when using ground truth peaks at high spatial resolution.

Supplementary Figure 6. Results of the high-resolution experiments performed directly on the ground truth vector field performed using deterministic streamline tractography (red and green) in conjunction with the scores of the original submissions (gray). We used two implementations of deterministic tractography methods: GT_1 is an in-house development of the Sherbrooke Connectivity Imaging Lab and GT_2 is implemented and openly available in MITK (www.mitk.org). Isotropic image resolution levels were 0.5mm, 0.75mm, 1.0mm, 1.25mm, 1.5mm, 1.75mm and 2.0mm. Overlap and overreach scores improved when compared to the original submissions. Regardless of the resolution, the number of invalid bundles remained above 80 for GT_1 and above 116 for GT_2 .

We implemented a further addition to the revised version of the manuscript: To allow future comparisons of algorithmic performance on higher quality high-resolution simulated datasets, we generated and made publicly available a new version of our phantom that replicates the quality of a Human Connectome Project (HCP) data set (cf. Supplementary Data 2).

The authors claim that their results are independent of data quality (by which they also mean spatial resolution) because one of their results includes using the “ground truth” orientations, but these have also been discretised, i.e. step (ii) has been applied (but not steps iii and iv). Evidently, had they used the original vector field (i.e. very high resolution), they would have recovered it in its entirety with zero error.

R2.6: A zero error might be what many would expect from tractography at high image quality and spatial resolution. However, hopefully we can convince the Reviewer with the additional experiments on high-resolution grids and the detailed arguments that we provided above that this will not be the case. We also hope that the simple example in Fig. 1 of the manuscript (global case) provides a convincing notion of why this is the case. The figure shows an example where two bundles pass a bottleneck, resulting in various tract hypotheses that differ from the hypothetical ground truth and cannot be sorted out at high resolution. Similar as well as much more complex situations are very common in the brain.

It follows that the claims of the paper are not general but dependent on the geometric and topological properties of the true continuous orientation field as well as on the resolution of the grid.

Any quantitative measure of error therefore is linked to these factors and cannot be trivially generalised. Anyone can design continuous orientation fields that challenge tractography to various degrees depending on how they are affected by discretisation. Error will change accordingly.

R2.7: The Reviewer is right: the exact numbers depend on geometry, complexity of this geometry and grid size. Nonetheless, it is the trend highlighted by the paper which we cannot escape. Local orientations that simultaneously represent multiple bundles will artificially create non-existent bundles. From what we know today, the more complex geometry of real brains can be expected to only worsen the problem.

During construction of the phantom, we aimed at representing those bundles of the brain that are well-known, and we even used tractography itself to generate the ground truth for the phantom. To the best of our knowledge, the design of the orientation field would not be expected to cause an over-exaggeration of false positives. It could cause a *positive* bias rather than a negative one, as discussed in the manuscript. The brain is much more complex than our phantom design. The phantom design might be much more complex than examples used in previous studies, but we are still far away from the brain's complexity.

The grid resolution in diffusion MRI will not solve the problem. Acquisition techniques have improved tremendously in the last years. However, increasing the resolution by a factor of 2 or 4 will not solve the problem or change the trend (see our novel experiments that we presented above and in the revised version of the manuscript, i.e. Supplementary Notes 3 and Supplementary Fig. 6).

Quantification and why it is biased

The approach taken by the authors to quantify errors in tractography is worth examining in detail since it is the basis for the main claim: that tractography is dominated by false positives.

Participants in the contest were asked to submit a set of candidate streamlines (continuous paths through the discrete orientation field). These are then compared to the simulated bundles of streamlines and classified as valid or invalid. Interestingly, the authors focus on the occurrences of valid vs invalid bundles (groups of streamlines) rather than the occurrences of valid vs invalid streamlines. They find that invalid bundles outnumber valid ones by a factor of 4, which is the basis of their main claim.

This is problematic for two reasons. First, invalid streamlines are grouped into so-called invalid bundles with a clustering algorithm, which ultimately determines their number. Unless these invalid bundles have the same properties as valid ones in terms of their shapes/lengths/widths/etc., there is no reason to expect them to have the same clustering properties. In fact, they are more likely to contain multiple clusters.

R2.8: We have analyzed the number streamlines distributions in the false-positive bundles (see histograms in Supplementary Fig. 5). The red line indicates the average distribution of percentile fiber counts in valid bundles, showing that the fiber counts of invalid bundles are similar to those of valid bundles. We used percentiles to account for the different total numbers of fibers submitted by the teams. The figure also visualizes examples of invalid bundles. From their appearance, it is not possible to differentiate them from valid bundles. They appear to be "real" in all aspects that the Reviewer mentions (shape/length/width etc.). For us, this makes perfect sense given the explanation for their existence: they are made up of parts of real bundles. This can be also nicely seen in Fig. 5c of the manuscript or Supplementary Fig. 5.

In the revised version of the manuscript, we extended our analysis to the distribution of streamlines between the invalid bundles as well between valid and invalid bundles in order to investigate this in

more depth (cf. Supplementary Fig. 8 and 9). Our analysis supports our previous results, meaning that many of the invalid bundles do not consist of spurious fibers but instead show the same properties as the valid bundles in terms of the number of streamlines and are by no means insignificant (see Supplementary Fig. 8 and Supplementary Fig. 9).

Supplementary Figure 8. Relative number of streamlines accumulated over the number of bundles sorted descending by their relative streamline count. The graphs show mean and standard deviation over all submissions, respectively (red: invalid, green: valid). A: On average, the first 10 valid bundles (VBs) (40%) contained 90% of all valid streamlines (vertical line). B: On average, the first 38 invalid bundles (IBs) (43%) contained 90% of all invalid streamlines (first vertical line, the second vertical line indicates the mean number of invalid bundles across all submissions). Thus, the invalid bundles did not consist of randomly distributed “stray” streamlines but instead were organized following a similar distribution as the valid bundles. The most obvious difference between the distributions was that the corpus callosum resembles a huge valid bundle with nearly 40% of all streamlines.

Supplementary Figure 9. Comparison of number of invalid bundles versus number of valid bundles at different fiber count thresholds. The threshold was selected relative to the total fiber count of the corresponding submission. It is not possible to get rid of the invalid bundles without losing a significant number of valid bundles. In fact, about half of the valid bundles is lost until the number of invalid bundles drops below the number of valid bundles. This implies, that the invalid bundles are by no means insignificant since their streamline count is similar to the streamline count of valid bundles.

Secondly, invalid bundles connect regions that the authors simulated as non-connected. There are 25 simulated connections (25 pairs of connected regions), and thus 275 possible wrong-connections,

outnumbering correct connections by a factor of 11. The authors report absolute numbers of valid/invalid bundles, but given the above, they should be reporting relative numbers, in which case their conclusions will change dramatically.

R2.9: We understand that relative numbers can provide another view of the problem. We now additionally report relative performance indices. There are two commonly used metric pairs qualifying for this task: precision/recall and sensitivity/specificity.

Sensitivity or recall (TP/P , $85\pm 15\%$) tell us whether we missed any relevant results (false negatives). Specificity (TN/N , $93\pm 5\%$) tells us whether we can trust tractography when it does *not* find one of the many theoretically possible connections. Precision ($TP/(TP + FP)$, $23\pm 9\%$) tells us whether the results are relevant (true positive) or spam (false positive).

The specificity is directly affected by the high number of theoretically possible connections that the Reviewer mentioned. Thus, despite the high invalid bundle counts, the specificity is still above 90%. So when tractography does *not* find a connection, in 93% of the cases this is a correct decision.

Looking at the comparison above, precision/recall best correspond to what we were interested in regarding tractography. This corresponds to the measures employed in other studies. We noticed that some tractography publications actually evaluate precision/recall but name these values sensitivity/specificity instead (e.g. Knösche et al., Hum Brain Mapp 2015, doi:10.1002/hbm.22902).

In the Supplementary Fig. 4 we depict an alternative to Fig. 3 in the manuscript that uses precision and recall instead of absolute numbers for the invalid bundles. Additionally, we show bundle overlap scores calculated with alternative metrics (precision and recall, Dice coefficient and Jaccard index). The figure yields a similar overall picture as its original version. We feel that the absolute numbers were more intuitive to understand and thus decided to keep the figure that we had in the original version of the manuscript.

The additionally reported relative performance metrics do not affect our conclusions regarding false positives, as they reveal the same strength and weaknesses of the methods and yield a similar distribution in the plot. A mean precision of 23% roughly corresponds to the 4:1 ratio of false- and true-positive bundles.

Supplementary Figure 4. This figure shows alternative metrics to the VB/IB and the OL/OR scores of Fig. 3 in the manuscript. Bundle detection rates are expressed in terms of precision and recall. To describe the bundle overlap, we calculated the Dice coefficient and Jaccard index in addition to precision and recall.

Interestingly, hidden deep into the supplementary material is the fact that by calculating the number of correct streamlines (as opposed to bundles), at least one of the entries scored a whopping 92% correct and 8% incorrect.

R2.10: The Reviewer would like to see the point more emphasized that the streamline-based analysis yielded high valid connection ratios for some teams. It is important to remember that even 100% valid connection ratio on streamline basis can be easily achieved for example by reconstructing only 1 streamline or by choosing the simplest bundle and recovering small parts of it. It can be extremely misleading to evaluate the valid connection ratio without considering the other metrics in context. This also applies to the mentioned entry, as we show in our answer to the following comment. Currently, some emphasis is given to the entry by marking it with a black arrow in Fig. 3a and using it an exemplary depiction of results in other figures. In our section “Selection of best-performing submissions” (Supplementary Notes 5) the entry is thoroughly discussed. Additionally, in the revised version of the manuscript, we added an explicit mentioning of this approach to the discussion (paragraph 3). In the alternative title that we proposed, we avoided ambiguities of the word

connection, which could have been interpreted mistakenly as “streamline” where it should rather be interpreted as “bundle”.

That is very far from what the authors conclude when they group results into bundles as they have done for their main results. In essence, what they have done is they have grouped the 8% incorrect streamlines in such a way that these 8% are clustered into a sufficiently large number of bundles that they now outnumber correct bundles by a factor of 4!

R2.11: The problem is, as mentioned in our answer to the previous comment, that the tractogram of this submission is quite empty (recall the trade-off). The percentage of correct streamlines is only useful if it is looked at in context. The corpus callosum on its own can make up a huge percentage of the streamlines submitted. Supplementary Fig. 8 and 9 in the revised version of the manuscript also show this quantitatively. To be more explicit: the 10 smallest valid bundles found by the mentioned submission make up for 2% of the valid connections, while the 10 smallest existing bundles in the ground truth make up for 14% of the existing connections. We discuss this now in the revised version of the manuscript (Supplementary Notes 5).

There are two other points to be made on the question of quantification. First, the major analyses are done as binary valid/invalid, whereas it is conceivable that some of the reconstructed connections might follow a correct route for a while before deviating from the correct trajectory.

R2.12: False positives are often made up of the “halves” or smaller parts of different valid bundles. In fact, a “deviation from the correct trajectory” is not even necessary to produce false connections. From a *local* point of view the trajectory can make perfect sense all the way along but the different parts could just belong to a different bundles (cf. Fig. 1 in the manuscript). Additional small deviations from the correct trajectory can as well be a huge problem when talking about connectivity. While an algorithm might have done the correct thing in all but one step during the tracking, the resulting connection could in consequence connect two regions that are not connected in reality. These problems and the ease with which we produce these problems as well as the corresponding trouble they cause can be considered an essential finding of the paper. It is also important to note that deviations from the correct trajectory are already tolerated by our distance-based identification of valid bundles (cf. Supplementary Fig. 13 for the results using traditional scoring without tolerated local deviations).

This is somewhat captured by their measure of overlap vs overreach, but is only considered in the cases of “valid bundles”.

R2.13: We allowed valid bundles with a certain “overreach” to relax our initial strict analysis that only allowed streamlines that: 1. Connected the right endpoints and 2. Never exited (“overreached”) the ground truth bundles volume (cf. Supplementary Fig. 12). If we would have kept the “strict” scoring the results in this paper would be disastrous for tractography (cf. Supplementary Fig. 13). We “accepted” that dMRI tractography is not a millimeter-exact technique and wanted to focus more on connectivity rather than expecting 100% geometric accuracy all along the path. The relaxation actually redefined invalid connections as valid. Considering a similar relaxation for invalid connections seems not useful. A certain notion of uncertainty was also provided for the invalid bundles: they were allowed to non-exactly connect their endpoints (majority voting between the streamlines was applied to choose what endpoints the bundle belongs to).

Secondly, there is no accounting for uncertainty associated with streamlines. There is a host of techniques for tractography that incorporate uncertainty, but the scoring system seems to ignore this. Although the authors claim that some of the submissions included probabilistic tractography methods, I don’t see how their scoring system accounts for the uncertainty information associated

with these methods, particularly once streamlines are grouped into bundles that are scored as a whole.

R2.14: If a probabilistic technique was used, the notion of uncertainty was handled by the submission, i.e. a user-defined uncertainty threshold was applied by the submitting group prior to submitting a set of streamlines. This information was now added to the Online Methods in the revised version of the manuscript.

One way to look more deeply into this issue of uncertainty using the current analysis pipeline would be a comparison of results obtained by varying uncertainty thresholds. It is correct that, in the future, a notion of weight or uncertainty per streamline could be included in the metrics. All this, however, would affect the trade-off between sensitivity and specificity, but not affect our conclusions with regard to the underlying problem.

MRI data

An important aspect of this paper is the very low quality of the data that have been shared with the contestants. In particular, the low number of direction (30) and relatively low b-value (1000 s/mm²) means that the deconvolution involved in undoing step (iii) of the simulation chain is particularly hard. While the authors claim that this choice was motivated by the fact that most data out there are of similarly low quality, the conclusions made in this paper (and the title itself) sound universal, and independent of data quality.

R2.15: We extended our experiments made on the *ground truth orientation field* in the revised version of the manuscript to additionally include *different levels of resolution* (Supplementary Fig. 6).

Another argument made by the authors is that spatial resolution is not a factor. I think it is an incredibly important factor: even in their ground-truth tractography where they have removed steps (iii) and (iv) of the processing, the authors still discretised the field. Their results therefore are dependent on the spatial resolution that they have chosen to use.

R2.16: We have discussed this extensively above. We agree that resolution is an interesting aspect that justifies more attention and additional experiments, which we now provide in the revised version of the manuscript. The results show that the findings remain valid and are not an artifact of the spatial resolution originally chosen.

Nowadays, MR technology enables ~1mm spatial resolution in vivo. It would be straightforward for the authors to simulate such data

R2.17: True. As already mentioned above, to allow future comparisons of algorithmic performance on higher quality high-resolution simulated datasets, we generated and made publicly available a new version of our phantom that replicates the quality of a Human Connectome Project (HCP) data set (Supplementary Data 2).

and assess the benefits in terms of reducing errors on particularly tricky brain connections.

R2.18: The focus on particularly tricky brain connections would indeed be an interesting analysis that is enabled using the presented methods. This could be a topic of future sub-challenges that focus on specific bundles that are relevant in particular applications of tractography.

Don't get me wrong

The arguments that I have put forward in this review may give the false impression that my view on tractography is filtered by pink goggles. It is not. I do think that diffusion tractography can be error

prone, but I also think that quantification of this error is very hard indeed. The authors focused on a very specific source of errors with tractography; I would say that it is arguably less important than other much more fundamental problems.

R2.19: In our view, given the broad application of tractography nowadays, the identified problems are already highly relevant to the field. This is despite “much more fundamental” problems that might as well exist, but are quite speculative at this point. As we pointed out in our discussion, we think that our findings will be important for the interpretation of tractography-based findings and will thus be very helpful to the community rather than damage it. Also, we see this paper as a starting point, a starting point that should be shared with the community and reacted upon with a series of follow-up papers.

Conclusion

This paper presents an interesting and valuable approach to the question of validating tractography. In addition, the general approach as well as the current framework and data will be an invaluable resource for the community to develop better methods in the future. However, I feel that the negative tone of this paper is both unjustified and potentially extremely harmful to the field.

R2.20: We put more effort in better balancing the report of our findings, trying to avoid a negative tone and equally highlight the positive aspects of tractography. We are convinced that publication of the findings will bring the field forward and not harm it: there are many examples of “blind” application of tractography in past papers, which is what we consider truly harmful for the field and maybe one reason why some people distrust tractography. We hope that we now managed to find the constructive tone required to support future applications of tractography and the interpretation of the respective results.

This paper does not quantify the occurrence of false positives in tractography, and therefore it cannot claim that tractography-based connectomes are dominated by false positives. At most, this paper shows that there are situations where false positives are likely to occur (to an unknown extent) due to lack of spatial resolution.

R2.21: Indeed, there are situations where false positives are likely to occur. As discussed above, the main cause is not trivial but more complex and in part related to the notion of connectivity and the fundamental problem formulation of dMRI tractography. From the knowledge that we now have, there is no reason to expect these false positives will disappear in real brain tractography (see Fig. 5 in the manuscript). Granted, we cannot quantify the exact numeric proportion in the real brain as we do in the phantom experiments, but for the reasons given above, we still expect real data tractograms to have a substantial amount of false positives. In any case, we now chose a toned down title of our paper and worked intensely on the balance of the report.

Written by: Saad Jbabdi

Reviewer #3 (Remarks to the Author):

In their paper entitled 'Tractography-based connectomes are dominated by false-positive connections' Maier-Hein et al. analysed the reproducibility of tractography across 20 research groups. While the method is ingenious,

Thank you!

my enthusiasm has been tempered by the lack of new findings.

In general:

Indeed, it is quite difficult to disprove findings obtained with tractography due to the lack of a gold standard model with regard to white matter anatomy in humans. While post-mortem validation studies do exist, they are sparse and technically difficult to achieve. Therefore, reproducibility of tractography findings remains an important factor.

However, here the authors circumvent the problems associated with the lack of such a model by ingeniously building a mock white matter gold standard based on the best tractography dissections available - as far as they know. They called their gold standard a 'synthetic ground truth' and subsequently derived a diffusion imaging dataset from this material. This dataset has been sent to 20 research groups for preprocessing and dissections in order to assess the reproducibility of the findings. Their results indicate a frightening low reproducibility across research groups, with 4 times more invalid bundles compared to the 'synthetic ground truth'.

Besides the ingenious approach described above, it is difficult to extract an important and productive scientific message from this paper. As it reads, the paper is focused on tractography caveats and does not offer a viable improved approach nor solution. Such a report may have a negative impact on the field of tractography by putting forth negative and undue influence on the use of methods available to researchers of white matter anatomy and function without providing any viable alternatives or solutions.

R3.1: We thank the Reviewer for openly addressing these concerns, which we address below.

It is true that our study does not provide a ready solution to the problems that we identified. However, we see our paper as an eye opener for future developments of better approaches. The findings are fundamental in their nature and provide an important starting point for future developments in the right direction. We present a unique validation technique together with validation results from a spectrum of different tractography methods, demonstrating the need for severe methodological innovation as opposed to fine tuning of existing methods.

Furthermore, and this is an important point to stress, we are convinced that researchers should, independent of the availability of alternatives, be aware of the caveats of their methods. Tractography is a great tool, and there are indeed currently no viable alternatives, but we must know what we can and what we cannot expect from this method.

In the revised version of the manuscript, we tried to better stress these points in the Discussion. The mentioned two points are the main reasons of why we are convinced that this report will have a positive impact on the field, and that it is our duty as scientists to publish these findings in a balanced but at the same time clear way.

There is a lack of novelty in the findings. The limitations of tractography have previously been demonstrated in many publications and books, particularly with regard to the organisation of complex fibres (such as Catani 2007, Jones 2008).

R3.2: In our experience from talking to people from the field or even contestants in our challenge, a-priori nobody would have expected such a result from the given setup. We also never expected this result. Only rigorous data crunching and endless looking at the massive number of submissions helped us unveiling the mechanisms underneath our unexpected result. The present study represents a systematic and quantitative identification and disentangling of the different potential sources of error, while previous studies were more observational and sometimes anecdotal. Only now that the paper is written and everything is summarized that people start realizing that these caveats indeed make sense, and that they do not relate to the given approach that is currently at use but rather to the underlying problem formulation in tractography. And they also start realizing the consequences for interpreting their own data, which is a very important effect. In response R1.6 above, we provide more thoughts regarding the novelty of our findings.

Additionally, the findings reported are not associated solely to the use of tractography but also to magnetic resonance imaging methods in general.

Tractography may have errors, but so does MR based cortical thickness and voxel based morphometry (Zilles et al. 2015) functional neuroimaging (Logthetis 2008) and T1 based myelin quantification (Sandrone et al. in press), voxel based lesion symptom mapping (Mah et al. Brain 2015). All of these approaches are limited because they assess the features of the living human brain based on an indirect magnetic resonance approach. Indirect measures are not exempt of errors.

R3.3: We agree with this statement; however, conceptually, we should not accept flaws in a method just because other methods also have flaws (which may or may not be related to the fact that the analyzed data were obtained through MRI; in the case of voxel based lesion symptom mapping, for instance, the main source of error is the use of univariate rather than multivariate analysis methods). It is certainly correct that the tractography algorithm itself is not the only source of error in the whole pipeline that starts with the magnetic resonance imaging experiment. Imaging is far from being error-free. Our study addresses two aspects related to this: 1) The question of how good tractography is if we imagine for a moment that the imaging method was perfect (see Supplementary Fig. 6 in the revised version of the Supplementary) and 2) the question of what magnitude of error we have to expect if we look at the whole pipeline including imaging, preprocessing and postprocessing.

More specifically:

The authors did not account for operator dependant errors, therefore shifting all the blame on tractography. This is indeed a limitation in their methods that should at least be mentioned.

R3.4: Thank you for raising this point, which is true when speaking about the actual challenge submissions. We did not assess whether the processing performed by the different teams was completely error-free. However, we addressed this by performing additional experiments with the most sophisticated tractography approaches available to us, namely a combination of global tractography with anatomical constraints (see Supplementary Fig. 7 in the revised version of the manuscript). We also checked what happens when we exclude all possible sources of error that we could think of (noise in the imaging, the modeling, etc., see Supplementary Fig. 6 in the revised version of the manuscript). This did not change the global picture that we had obtained by looking at the challenge submissions. As suggested by the Reviewer, in the revised version of the manuscript we now mention this point in the Online Methods of the revised version of the manuscript.

The text indicates “that Some of these false-positive bundles resemble previously reported pathways identified by in-vivo tractography, such as the frontal aslant tractor the vertical occipital fasciculus” and later states “The existence of the FAT, SFOF and VOF is controversial (41,42,49,54)”. Do the

authors suggest that these tracts do not exist in humans? This statement requires clarification because it constitutes a direct attack on previous work, being mindful that the FAT and SFOF have been validated with post-mortem dissection. Disproving these findings with a 'synthetic ground truth' will challenge the credibility of the rest of their findings.

R3.5: Thank you for raising this point. We now removed the statement from the introduction and only raise this point later accompanied with an explicit clarification. We did not intend to suggest that these tracts are absent in the human brain. One important conclusion from our study must be, however, that the evidence for the existence of any tract should never be taken solely from tractography at its current state.

In conclusion, we have addressed all points raised by the Reviewers, and we hope that the revised paper is now fit for publication in Nature Communications.

Reviewers' comments:

Reviewer #1 (Remarks to the Author):

The real efforts of the authors to tone down their claims must be acknowledged, as well as their clear willing to provide detailed argued answers. However, I still consider that Nature Communication is not the adequate journal for such a paper that summarizes the results obtained in the frame of a conference challenge. The level of novelty of this work does not match the level of novelty required to publish in such a journal. A journal focused on methods would be more appropriate. Again, the general message is not novel but aggregates a collection of knowledge that was already well-known by experts of the field, although not quantified. While the paper provides some interesting quantification of tractography errors with respect to the plethora of pipelines used by the co-authors, it does not provide the solution to prevent them, but only recommendations. Several solutions to the ill-posed nature of tractography are already under investigation in the diffusion community and will probably help fighting against some of the issues, but they won't be available to users before a couple of years.

Reviewer #2 (Remarks to the Author):

I would like to thank the authors for addressing my comments, and in particular, for the significant change of tone overall in the paper, as exemplified in the change of title. Thank you.

I am happy with the paper as it is now. The only remaining comment I have is I still feel that the 4:1 ratio of invalid versus valid bundles is presented in a negative light. As I said in my previous review, since there are more than 10 times more possible invalid than there are valid bundles, this ratio of 4:1 is actually good news for tractography (although the 10:1 ratio does not relate in a straightforward way to a null because of smoothness in the data/tractography process). Also this ratio of 4:1 depends on how many bottlenecks there are in the simulations, so it should be emphasised that it does not mean that any one who is doing tractography should expect that 4:1 ratio in their experiment.

Other than that, I think it is a good paper that should hopefully get people thinking.

Reviewer #3 (Remarks to the Author):

Although the authors made some effort to edit their manuscript, the overall message is still unclear. This is mainly coming from some provocative statements that need to be changed in order to fit with the present findings as well as the recent revision that do not respect the logical flow of the manuscript.

The main message of the study is that tractography is appropriate to reconstruct in the living brain white matter anatomical features previously known from post-mortem dissections but not fit to rewrite white matter neuroanatomy.

However in it's current state the manuscript does not convey this message clearly. Therefore, here are some edits I would recommend in order to convey the correct message to the community:

Starting from the title:

"Tractography-based connectomes are dominated by false-positive connections." to be changed for "Data driven tractography-based connectomes are dominated by false-positive connections"

"An encouraging finding is that most state-of-the-art algorithms reconstructed 90% of ground truth bundles to at least some extent" to be changed for "An encouraging finding is that most state-of-the-art algorithms combined with manual or automatic anatomically informed white matter dissection reconstructed 90% of ground truth bundles to at least some extent"

"On the other hand, the algorithms produced four times more invalid than valid bundles, and half of these invalid bundles occurred systematically in the majority of submissions." to be changed for "On the other hand, the same algorithms produced data driven connectome including four times more invalid than valid bundles, and half of these invalid bundles occurred systematically in the majority of submissions."

"novel framework for methodological validation" The term validation is misleading, since the present approach does not confirm findings are anatomically correct. I would rather recommend the authors to use the following "novel framework for tractography reliability estimation"

"Tractography identified majority of existing bundles" to be changed for "anatomical dissections of tractography identified majority of existing bundles"

"Tractography identified more invalid than valid bundles" to be changed for "data driven connectomes identified more invalid than valid bundles"

"While the existence of the FAT, SFOF and VOF is controversial^{46,50,55,56}," This is incorrect since 3 out of the 4 studies cited are post-mortem validation of these findings.

but the rest of the sentence is very important "the presented findings only show that evidence for the existence of tracts should not be taken solely from tractography at its current state."

"Tractography is fundamentally ill-posed" I guess what the authors really mean here is

“Limitation and future challenges for tractography”

“Fundamentally, tractography will require severe methodological innovation to become tractable” this does not make sense “Fundamentally, there is an urgent need for methodological innovation in tractography in order to build an anatomically correct human connectome”

I would overall encourage the authors to take some distance from their findings in order to rewrite their manuscript entirely in light of these comments.

Reviewers' comments:

Reviewer #1 (Remarks to the Author):

The real efforts of the authors to tone down their claims must be acknowledged, as well as their clear willing to provide detailed argued answers. However, I still consider that Nature Communication is not the adequate journal for such a paper that summarizes the results obtained in the frame of a conference challenge. The level of novelty of this work does not match the level of novelty required to publish in such a journal. A journal focused on methods would be more appropriate. Again, the general message is not novel but aggregates a collection of knowledge that was already well-known by experts of the field, although not quantified. While the paper provides some interesting quantification of tractography errors with respect to the plethora of pipelines used by the co-authors, it does not provide the solution to prevent them, but only recommendations. Several solutions to the ill-posed nature of tractography are already under investigation in the diffusion community and will probably help fighting against some of the issues, but they won't be available to users before a couple of years.

We thank the Reviewer for acknowledging our effort in revising the manuscript and toning down the claims. We hope that our work will attract interest and guide method developers into the right direction to innovate tractography.

Reviewer #2 (Remarks to the Author):

I would like to thank the authors for addressing my comments, and in particular, for the significant change of tone overall in the paper, as exemplified in the change of title. Thank you.

I am happy with the paper as it is now. The only remaining comment I have is I still feel that the 4:1 ratio of invalid versus valid bundles is presented in a negative light. As I said in my previous review, since there are more than 10 times more possible invalid than there are valid bundles, this ratio of 4:1 is actually good news for tractography (although the 10:1 ratio does not relate in a straightforward way to a null because of smoothness in the data/tractography process). Also this ratio of 4:1 depends on how many bottlenecks there are in the simulations, so it should be emphasised that it does not mean that any one who is doing tractography should expect that 4:1 ratio in their experiment.

Other than that, I think it is a good paper that should hopefully get people thinking.

We thank the Reviewer for the positive summary of our work and the acknowledgment our efforts in improving the manuscript. We have revised the paper to emphasize that quantitative numbers and ratios reported in this work only apply to the generated numerical phantom and should not be blindly transferred to tractography on real human brain data. The following wording is used:

“The employed simulation-based approach cannot quantify the effects related to in vivo connectivity in an absolute sense; that is, our results do not mean that anyone who is doing tractography should expect the reported VB-to-IB and coverage-to-overreach ratios”

We have also removed the 4:1 ratio from the abstract, results, discussion and toned down the message further to account for your comment.

Reviewer #3 (Remarks to the Author):

Although the authors made some effort to edit their manuscript, the overall message is still unclear. This is mainly coming from some provocative statements that need to be changed in order to fit with the present findings as well as the recent revision that do not respect the logical flow of the manuscript.

The main message of the study is that tractography is appropriate to reconstruct in the living brain white matter anatomical features previously known from post-mortem dissections but not fit to rewrite white matter neuroanatomy.

However in it's current state the manuscript does not convey this message clearly. Therefore, here are some edits I would recommend in order to convey the correct message to the community:

We thank the Reviewer for acknowledging our effort in improving the manuscript. Your summary of the main message, in your own terms, is appropriate. However, it hides other important messages that are:

1. Tractography can reconstruct the known white matter bundles, but only to a partial extent, and the precise origins and terminations of these bundles are still an open question. Only the core of such bundles can be reliability reconstructed.
2. False positives are present, prominent and cannot be removed by thresholding based on size or tract count.

Overall, we have implemented all your suggestions for edits. We believe that the findings are now stated much more clearly.

Starting from the title:

“Tractography-based connectomes are dominated by false-positive connections.” to be changed for “Data driven tractography-based connectomes are dominated by false-positive connections”

The comment refers to the previous title of the paper. We have a new, toned down title that the other two reviewers liked. So, we would keep the following title:

The challenge of mapping the human connectome based on diffusion tractography

“An encouraging finding is that most state-of-the-art algorithms reconstructed 90% of ground truth bundles to at least some extent” to be changed for “An encouraging finding is that most state-of-the-art algorithms combined with manual or automatic anatomically informed white matter dissection reconstructed 90% of ground truth bundles to at least some extent”

Thank you for the suggestion. We agree that our statement was potentially misleading since the algorithms did not directly reconstruct bundles but only complete tractograms. Because “manual or automatic anatomically informed white matter dissection” does not apply to most of the submissions, we came up with a new sentence: “An encouraging finding is that most state-of-the-art algorithms produced tractograms with 90% of the ground truth bundles contained to at least some extent.”

“On the other hand, the algorithms produced four times more invalid than valid bundles, and half of these invalid bundles occurred systematically in the majority of submissions.” to be changed for “On the other hand, the same algorithms produced data driven connectome including four times more invalid than valid bundles, and half of these invalid bundles occurred systematically in the majority of submissions.”

Again, thank you for the suggestion. We tried to avoid the word “connectome” when referring to the submissions, since teams produced tractograms without reference to any ending or connecting regions. Also, we would like to keep a similar wording as for the valid bundles above, since both are based on the same input (full brain tractograms) using the same scoring system that identified VBs and IBs. Thus, we propose the following modification: “On the other hand, the same tractograms contained many more invalid than valid bundles, and half of these invalid bundles occurred systematically in the majority of submissions.”.

“novel framework for methodological validation” The term validation is misleading, since the present

approach does not confirm findings are anatomically correct. I would rather recommend the authors to use the following 'novel framework for tractography reliability estimation'

We fully agree with you. Validation is a strong term. We have implemented your suggestion everywhere in the manuscript.

"Tractography identified majority of existing bundles" to be changed for "anatomical dissections of tractography identified majority of existing bundles"

"Tractography identified more invalid than valid bundles" to be changed for "data driven connectomes identified more invalid than valid bundles"

Again, thank you for these suggestions. We have modified the section headings following the wording that we introduced in response to your comments above.

"While the existence of the FAT, SFOF and VOF is controversial^{46,50,55,56}," This is incorrect since 3 out of the 4 studies cited are post-mortem validation of these findings. but the rest of the sentence is very important "the presented findings only show that evidence for the existence of tracts should not be taken solely from tractography at its current state."

You are completely right. We have removed that part of the sentence and kept the last part. Thank you for pointing this out.

"Tractography is fundamentally ill-posed" I guess what the authors really mean here is "Limitation and future challenges for tractography"

Yes, that is what we mean. We have done this modification.

"Fundamentally, tractography will require severe methodological innovation to become tractable" this does not make sense "Fundamentally, there is an urgent need for methodological innovation in tractography in order to build an anatomically correct human connectome"

We have rephrased this sentence as suggested.

I would overall encourage the authors to take some distance from their findings in order to rewrite their manuscript entirely in light of these comments.

A co-author that had distance from the paper, due to a different background, went through the manuscript, taking into account your comments and those of the other Reviewers. While we have not re-written the paper entirely, we have clarified the messages according to your suggestions. Thank you very much for your helpful suggestions.

Once again, we thank you for your constructive feedback and support. We also thank the Reviewers for their stimulating comments. They truly contributed in improving the paper.